# Temporal In-Context Fine-Tuning with Temporal Reasoning for Versatile Control of Video Diffusion Models

**Kinam Kim*  Junha Hyung*  Jaegul Choo**
KAIST AI
{kinamplify, sharpeeee, jchoo}@kaist.ac.kr

## Abstract

Recent advances in text-to-video diffusion models have enabled high-quality video synthesis, but controllable generation remains challenging—particularly under limited data and compute. Existing fine-tuning methods for conditional generation often rely on external encoders or architectural modifications, which demand large datasets and are typically restricted to spatially aligned conditioning, limiting flexibility and scalability. In this work, we introduce Temporal In-Context Fine-Tuning (TIC-FT), an efficient and versatile approach with temporal reasoning for adapting pretrained video diffusion models to diverse conditional generation tasks. Our key idea is to concatenate condition and target frames along the temporal axis and insert intermediate *buffer frames* with progressively increasing noise levels. These buffer frames enable smooth transitions, aligning the fine-tuning process with the pretrained model's temporal dynamics. TIC-FT is architecture-agnostic and achieves strong performance with as few as 10–30 training samples. We validate our method across a range of tasks—including image-to-video and video-to-video generation—using large-scale base models such as CogVideoX-5B and Wan-14B. Extensive experiments show that TIC-FT outperforms existing baselines in both condition fidelity and visual quality, while remaining highly efficient in both training and inference. For additional results, visit `https://kinam0252.github.io/TIC-FT/`.

## 1  Introduction

Text-to-video generation models have advanced rapidly, reaching quality levels suitable for professional applications [1, 2, 3, 4, 5, 6]. Beyond basic generation, recent research has increasingly focused on leveraging pretrained models to enable more precise control and conditional guidance, addressing the growing demand for finer adjustments and more nuanced generation capabilities [7, 8, 9, 10, 11, 12, 13, 14, 15, 16].

Despite this progress, current fine-tuning approaches for conditioning video diffusion models face notable limitations. Many methods require large training datasets and introduce additional architectural components, such as ControlNet [7] or other external modules, which impose substantial memory overhead. Moreover, the reliance on external encoders for conditioning often leads to the loss of fine-grained details during the encoding process. ControlNet-style methods [16, 17, 14], in particular, operate within rigid conditioning frameworks: they are primarily designed for spatially aligned conditions and require conditioning signals to match the target video length. For example, when conditioning on a single image, common workarounds include replicating the image across the temporal dimension to align with the video frames or embedding it as a global feature. These

---

* indicates equal contribution.

39th Conference on Neural Information Processing Systems (NeurIPS 2025).

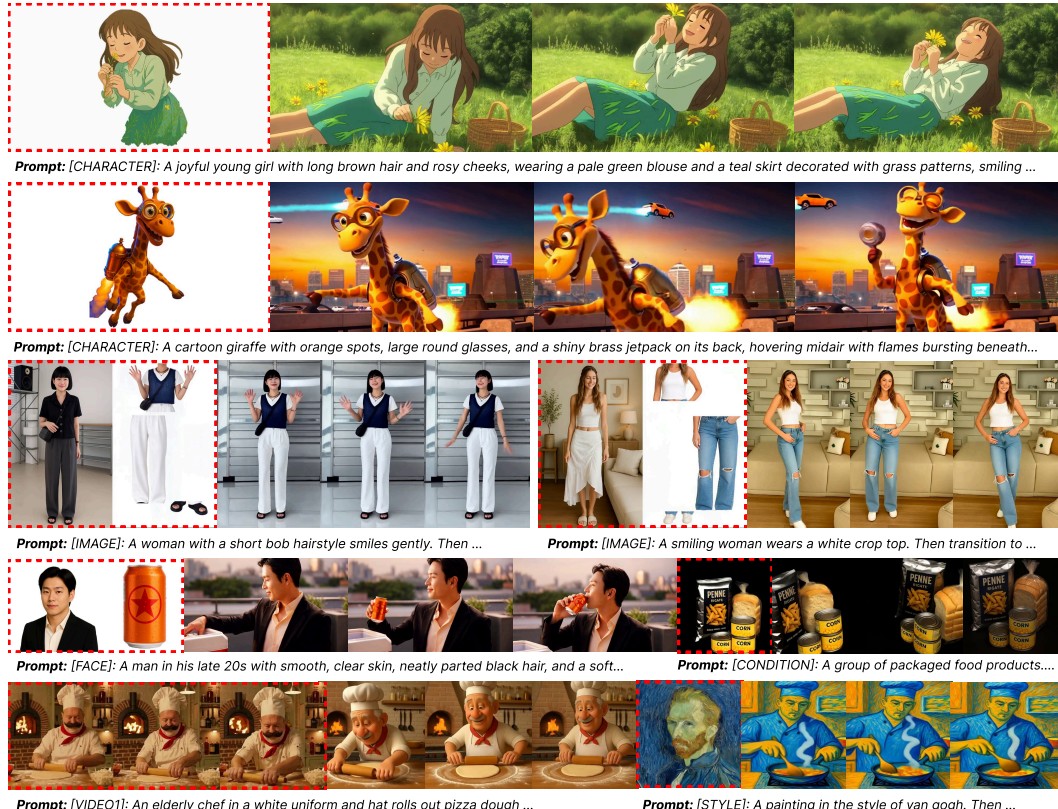

Figure 1: Demonstration of our method across diverse tasks, including character-to-video, virtual try-on, ad-video generation, object-to-motion, toonification, and video style transfer.

approaches typically necessitate task-specific adaptations of the conditioning pipeline. Alternative fine-tuning strategies, such as IP-Adapter [18] and latent concatenation [10], encounter similar challenges regarding flexibility and computational cost, as they modify or expand the pretrained model architectures.

In contrast, in-context learning (ICL) [19] offers a more efficient and versatile paradigm. ICL is training-free and can be flexibly applied to user-defined tasks by providing examples directly within the input context, eliminating the need for additional parameter updates. While ICL has shown strong success in large language models [19, 20], its application to image and video generation has primarily been explored in autoregressive models [21, 22], with limited adaptation to diffusion models.

Efforts to implement ICL in diffusion models [23, 15] have often relied on ControlNet-style training approaches, which contradict the core advantage of ICL: leveraging pretrained distributions without additional training. Departing slightly from the pure ICL paradigm, recent work has introduced in-context LoRA [8], a related technique that enables consistent image generation by producing multiple images in a single forward pass arranged in grids, thereby facilitating information sharing across images. With minimal fine-tuning, this method achieves high-quality and highly consistent results, benefiting from the inherent in-context generation capabilities of pretrained text-to-image models, which are naturally suited for grid-based generation.

In contrast, video generation models possess far less of this capability. Although concurrent research has explored extending in-context LoRA to video generation [24], these models are poorly suited to producing grid-like outputs, making the approach significantly more training-intensive and less effective. Furthermore, these methods are not inherently designed for conditional generation and often depend on training-free inpainting strategies [25, 26], which tend to degrade performance. They also lack flexibility in handling mismatches between the condition length and the number of target frames, as there are no straightforward solutions for general cases. In the simple case of conditioning on a single image, the image must be redundantly replicated across all frames, resulting in substantial increases in memory usage and computational overhead.

In this paper, we propose a highly effective and versatile fine-tuning method for conditional video diffusion models: temporal in-context fine-tuning. Instead of spatially concatenating condition and target inputs, our approach aligns them temporally—concatenating condition frame(s) and target frame(s) along the time axis—and fine-tunes the model using only a minimal number of samples. This design leverages the inherent capability of pretrained video diffusion models to process temporally ordered inputs, enabling effective generation when condition and target frames are arranged sequentially.

To ensure a smooth transition between the condition and target frames, we introduce buffer frames—intermediate frames with monotonically increasing noise levels that bridge the gap between the clean condition frames and the fully noised target frames. These buffer frames facilitate smooth, natural fade-out transitions from condition to generated frames, preventing abrupt scene transitions and preserving consistency with the pretrained model's distribution. Combined with this design, our method enables fine-tuning with as few as 10–30 training samples. Additionally, our method preserves the original model architecture without introducing additional modules, thereby reducing VRAM requirements.

The proposed approach also allows the model to leverage condition frames directly through unified 3D attention, avoiding the detail loss typically introduced by external encoders. Furthermore, it enables versatile conditional generation by eliminating the need for spatial alignment and accommodating a wide range of condition lengths—from single images to full video sequences—thereby supporting diverse video-to-video translations and image-to-video generation tasks.

**In summary, our main contributions are as follows:**

- We propose *temporal in-context fine-tuning*, a simple yet highly effective method for conditional video diffusion that minimizes the distribution mismatch between pretraining and fine-tuning, without requiring architectural modifications.

- We demonstrate strong performance with minimal training data (10–30 samples), offering a highly efficient fine-tuning strategy.

- Our method enables versatile conditioning, supporting variable-length inputs and unifying diverse image- and video-conditioned generation tasks within a single framework.

- We validate our method across a wide range of tasks, including reference-to-video generation, motion transfer, keyframe interpolation, and style transfer with varying condition content and lengths.

## 2 Related work

**Conditional Video Diffusion Models.** Many conditional video generation methods [7, 8, 9, 10, 11, 12, 13, 14, 15, 16] rely on auxiliary encoders (e.g., ControlNet [7]) or architectural modifications (e.g., IP-Adapter [18]), which prevent full exploitation of the pretrained model's capabilities. These approaches typically require larger datasets, longer training, and incur significant memory overhead. Moreover, they are often limited to spatially aligned conditioning, making them less suitable for variable-length or misaligned condition–target pairs.

**In-Context Learning for Diffusion Models.** Inspired by its success in language models [19, 20], in-context finetuning (IC-FT) has been explored in visual domains via grid-based generation [8, 24], but its extension to video is limited. Videos rarely follow grid layouts, and inference methods like SDEdit [25] degrade output quality. Moreover, these approaches assume strict condition–output alignment, making them unsuitable for flexible conditional video generation.

**Diffusion with Heterogeneous Noise Levels.** Recent works such as *FIFO-Diffusion*[27] and *Diffusion Forcing*[28] demonstrate that diffusion models can effectively operate on sequences with varying noise levels across frames or tokens—challenging the conventional assumption of uniform noise and motivating our use of buffer frames with progressively increasing noise.

Building on these ideas, we propose **Temporal In-Context Fine-Tuning (TIC-FT)**—a simple yet effective method that temporally concatenates condition and target frames, inserting buffer frames with increasing noise levels to smooth abrupt transitions in both scene content and noise levels. Unlike

ControlNet-style methods, TIC-FT is architecture-agnostic and naturally supports variable-length, spatially misaligned condition–target pairs.

## 3 Method

### 3.1 Preliminaries

We briefly review diffusion-based text-to-video generation. A video with $F_{\text{in}}$ RGB frames is $\mathbf{x}_{1:F_{\text{in}}} \in \mathbb{R}^{F_{\text{in}} \times 3 \times H_{\text{pix}} \times W_{\text{pix}}}$. A spatio-temporal encoder $\phi$ maps it to latents $\mathbf{z}^{(0)} = \mathbf{z}^{(0)}_{1:F} = \phi(\mathbf{x}_{1:F_{\text{in}}}) \in \mathbb{R}^{F \times C \times H \times W}$ with $F \leq F_{\text{in}}$, $H \leq H_{\text{pix}}$, and $W \leq W_{\text{pix}}$, and a decoder $\psi$ approximately inverts $\phi$. Latent frames are diffused by $q(\mathbf{z}^{(0)}, t) := \mathbf{z}^{(t)} = \alpha_t \mathbf{z}^{(0)} + \sigma_t \varepsilon$ for $t \in \{0, \dots, T\}$ and $\varepsilon \sim \mathcal{N}(\mathbf{0}, \mathbf{I})$, with a predefined schedule $(\alpha_t, \sigma_t)$. A DiT[29] backbone $\epsilon_\theta$ predicts the noise and is trained with

$$\mathcal{L}_{\text{diff}} = \mathbb{E}_{\mathbf{z}^{(0)}, \mathbf{c}, \varepsilon, t}\left[\left\|\varepsilon - \epsilon_\theta(\mathbf{z}^{(t)}, t, \mathbf{c})\right\|_2^2\right], \tag{1}$$

where latent frames and paired text condition $\mathbf{c}$ is sampled from the dataset. Generation starts from $\mathbf{z}^{(T)} \sim \mathcal{N}(\mathbf{0}, \mathbf{I})$ and iteratively applies a sampler $\mathbf{z}^{(t-1)} = \mathcal{S}(\mathbf{z}^{(t)}, t, \mathbf{c}; \epsilon_\theta)$ until $\mathbf{z}^{(0)}$, which $\psi$ decodes to video.

### 3.2 Temporal concatenation

**Overview** We introduce overall pipeline of the proposed *temporal in-context fine-tuning* (TIC-FT) in this section. We first detail the temporal concatenation of condition and target latents- with **buffer frames** that ease the abrupt scene and noise-level transition—followed by the **inference** and **training** procedures formalized in Algorithms 1–2.

**Setup.** The task is to generate a sequence of target frames of length $K$, denoted as $\hat{\mathbf{z}}^{(0)} = [\hat{\mathbf{z}}^{(0)}_{L+1}, \dots \hat{\mathbf{z}}^{(0)}_{L+K}]$, conditioned on a set of input frame(s): $\bar{\mathbf{z}}^{(0)} = [\bar{\mathbf{z}}^{(0)}_1, \dots \bar{\mathbf{z}}^{(0)}_L]$. Our approach concatenates the condtion and target frames along the temporal axis. A naïve formulation simply places the clean condition frames directly before the noisy target frames:

$$\mathbf{z}^{(t)} = \underbrace{\bar{\mathbf{z}}^{(0)}_{1:L}}_{\text{condition}} \parallel \underbrace{\hat{\mathbf{z}}^{(t)}_{L+1:L+K}}_{\text{target}} \quad \in \mathbb{R}^{(L+K) \times C \times H \times W}. \tag{2}$$

Here, $\hat{\mathbf{z}}^{(t)}_{L+1:L+K}$ represents the target latent frames at denoising timestep $t$.

At inference time, we initialize with $\bar{\mathbf{z}}^{(0)} \parallel \hat{\mathbf{z}}^{(T)}$ and iteratively denoise the concatenated frames with

$$\bar{\mathbf{z}}^{(0)} \parallel \hat{\mathbf{z}}^{(t-1)} = \mathcal{S}(\bar{\mathbf{z}}^{(0)} \parallel \hat{\mathbf{z}}^{(t)}, t, \mathbf{c}; \epsilon_\theta) \tag{3}$$

until reaching $\mathbf{z}^{(0)} = \bar{\mathbf{z}}^{(0)} \parallel \hat{\mathbf{z}}^{(0)}$. At each denoising step, only the $K$ target frames are denoised, while the condition frames are fixed to enforce consistency. The final output video corresponds to the target slice $\mathbf{z}^{(0)}_{L+1:L+K}$.

The flexibility of varying $L$ allows this formulation to generalize across a wide range of conditional video generation tasks. When $L = 1$, the problem becomes an *image-to-video* generation task: producing a full video sequence from a single reference image together with a text description of the sequence.

### 3.3 Buffer frames

Unlike conventional image-to-video (I2V) approaches, where the condition acts as the first frame of the output, our setup also allows for discontinuous conditioning, broadening its applicability. For $L > 1$, the method naturally extends to *video-to-video* generation. A reference clip can perform *video style transfer* by transferring its appearance onto a new motion sequence. Likewise, providing an action snippet along with a query frame enables *in-context action transfer*, where the observed motion is adapted to a novel scene. Supplying sparsely sampled frames supports *keyframe interpolation*, allowing the model to smoothly generate intermediate transitions between distant frames.

Thus, simple temporal concatenation serves as a unified and highly versatile framework for diverse conditional video generation tasks.

However, this naïve approach is suboptimal for fully leveraging the capabilities of the pretrained video diffusion model. Aligning the finetuning task as closely as possible with the pretrained model's distribution is essential to achieve high efficiency—enabling strong performance with minimal data and computational resources. Thus, it is desirable to design the finetuning process around tasks the model is already proficient at.

Direct concatenation violates this principle in two key ways. First, in scenarios where the target frames do not naturally continue from the condition frames—i.e., when there is an abrupt scene transition between the last condition frame and the first target frame—the model is forced to synthesize highly discontinuous content. Pretrained video diffusion models are typically trained on smoothly evolving sequences and lack the inherent capability to handle such abrupt transitions, as datasets with sudden scene changes are commonly filtered out during data curation. Second, diffusion models are not designed to denoise sequences containing frames with heterogeneous noise levels, as would occur when combining clean condition frames with noisy target frames during the sampling process.

We therefore introduce $B$ intermediate buffer frames that perform **temporal reasoning**, whose noise levels $\tilde{\tau}_b$ linearly bridge $0$ and $T$:

$$\tilde{\mathbf{z}}^{(\tilde{\tau}_{1:B})} = \big[\, \tilde{\mathbf{z}}_1^{(\tilde{\tau}_1)}, \ldots, \tilde{\mathbf{z}}_B^{(\tilde{\tau}_B)} \big], \qquad \tilde{\tau}_b = \frac{b}{B+1}T. \tag{4}$$

There can be different design choices for the buffer frames, and we empirically find that using the noised condition frames, $\tilde{\mathbf{z}}^{(t)} = \bar{\mathbf{z}}^{(t)}$, yields a good performance. Then the full initial latent sequence becomes

$$\mathbf{z}^{(T)} = \underbrace{\bar{\mathbf{z}}_{1:L}^{(0)}}_{\text{condition}} \,\|\, \underbrace{\tilde{\mathbf{z}}_{L+1:L+B}^{(\tilde{\tau}_{1:B})}}_{\text{buffer}} \,\|\, \underbrace{\hat{\mathbf{z}}_{L+B+1:L+B+K}^{(T)}}_{\text{target}} . \tag{5}$$

## 3.4   Inference

Let $\mathcal{T}(\mathbf{z}^{(t)})$ be a noise level list corresponding to the latent sequence $\mathbf{z}^{(t)}$: $\mathcal{T} : \mathbb{R}^{F \times C \times H \times W} \longrightarrow \{0, \ldots, T\}^F$. The initial noise level list at $t = T$ is

$$\mathcal{T}(\mathbf{z}^{(T)}) = \big[\, 0, \tilde{\tau}_1, \ldots, \tilde{\tau}_B, T, \ldots, T \big] \in \{0, \ldots, T\}^{L+B+K}. \tag{6}$$

At any global timestep $t$, we define the noise levels as:

$$\mathcal{T}(\mathbf{z}^{(t)}) = \big[\, 0, \tau_1(t), \ldots, \tau_B(t), t, \ldots, t \big], \tag{7}$$

where $\tau_b(t) = \tilde{\tau}_b$ if $\tilde{\tau}_b < t$, and $\tau_b(t) = t$ otherwise.

Our inference algorithm proceeds by iteratively identifying the frames currently at the maximal noise level $t$ and applying the video diffusion sampler exclusively to those frames. This process continues from $t = T$ down to $t = 0$. The full inference procedure is detailed in Algorithm 1.

## 3.5   Training

For each video–text pair $(\bar{\mathbf{z}}^{(0)}, \hat{\mathbf{z}}^{(0)}, \mathbf{c}) \sim \mathcal{D}$, the training proceeds as follows. First, we randomly sample a global timestep $t \sim \mathcal{U}\{1, \ldots, T\}$ and Gaussian noise $\varepsilon \sim \mathcal{N}(\mathbf{0}, \mathbf{I})$. Next, we construct the noised model input sequence $\mathbf{z}^{(t)}$ with the noise level defined in Eq. 7.

The model then predicts the noise $\hat{\varepsilon} = \epsilon_\theta(\mathbf{z}^{(t)}, t, \mathbf{c})$ for all frames. However, the loss is computed only over the target frames to avoid enforcing supervision for the buffer frames. Specifically, we minimize the mean squared error between the true noise and the predicted noise over the target frame indices, defined as $\mathcal{L} = \frac{1}{K} \sum_{i=L+B+1}^{L+B+K} \big\| \varepsilon_i - \hat{\varepsilon}_i \big\|_2^2$. The model parameters $\theta$ are updated via a gradient step computed from this loss. By excluding the buffer frames from the loss calculation, the network is free to predict whatever is most natural for these frames, thereby preventing spurious gradients that could shift the model away from the pretraining distribution. In practice, we observe that the buffer frames often evolve into a smooth fade-out and fade-in transition between the condition and target frames. The full training procedure is summarized in Algorithm 2.

---
**Algorithm 1:** TIC-FT inference
---

**Input:** Clean condition latents $\bar{\mathbf{z}}^{(0)}$; buffer noise levels $\tilde{\tau}_{1:B}$; text prompt $\mathbf{c}$; denoiser $\epsilon_\theta$
**Output:** Denoised target latents $\hat{\mathbf{z}}^{(0)} = \mathbf{z}_{L+B+1:L+B+K}^{(0)}$

Generate buffer latents $\tilde{\mathbf{z}}^{(\tilde{\tau}_{1:B})} = q\big(\bar{\mathbf{z}}^{(0)}, \tilde{\tau}_{1:B}\big)$;             // add noise
Sample target latents $\hat{\mathbf{z}}^{(T)} \sim \mathcal{N}(\mathbf{0}, \mathbf{I})$;
Concatenate $\mathbf{z}^{(T)} \leftarrow \bar{\mathbf{z}}^{(0)} \,\|\, \tilde{\mathbf{z}}^{(\tilde{\tau}_{1:B})} \,\|\, \hat{\mathbf{z}}^{(T)}$;
**for** $t = T$ **to** $1$ **do**             // global time descending
    $\mathbf{t} \leftarrow \mathcal{T}\big(\mathbf{z}^{(t)}\big)$;              // noise-level vector
    $\mathcal{A} \leftarrow \{\, i \mid \mathbf{t}_i = t \,\}$;
    $\mathbf{z}_{\mathcal{A}}^{(t-1)} \leftarrow \mathcal{S}\big(\mathbf{z}^{(t)}, t, \mathbf{c}; \epsilon_\theta\big)_{\mathcal{A}}$;

**return** $\mathbf{z}_{L+B+1:L+B+K}^{(0)}$

---

---
**Algorithm 2:** TIC-FT training
---

**Input:** Dataset $\mathcal{D}$ with tuples $(\bar{\mathbf{z}}^{(0)}, \hat{\mathbf{z}}^{(0)}, \mathbf{c})$; buffer levels $\tilde{\tau}_{1:B}$; noise schedule $(\alpha_t, \sigma_t)$
**Output:** Fine-tuned parameters $\theta$

**foreach** minibatch $(\bar{\mathbf{z}}^{(0)}, \hat{\mathbf{z}}^{(0)}, \mathbf{c}) \sim \mathcal{D}$ **do**
    **foreach** sample in minibatch **do**
        Sample $t \sim \mathcal{U}\{1, \dots, T\}$ and $\varepsilon \sim \mathcal{N}(\mathbf{0}, \mathbf{I})$;
        $\tilde{\mathbf{z}}^{(\tau_{1:B}(t))} \leftarrow q\big(\bar{\mathbf{z}}^{(0)}, \tau_{1:B}(t)\big)$;
        $\hat{\mathbf{z}}^{(t)} \leftarrow \alpha_t \hat{\mathbf{z}}^{(0)} + \sigma_t \varepsilon$;
        $\mathbf{z}^{(t)} \leftarrow \bar{\mathbf{z}}^{(0)} \| \tilde{\mathbf{z}}^{(\tau_{1:B}(t))} \| \hat{\mathbf{z}}^{(t)}$;
        $\hat{\varepsilon} \leftarrow \epsilon_\theta(\mathbf{z}^{(t)}, t, \mathbf{c})$;
        $\mathcal{L} \leftarrow \frac{1}{K} \sum_{i=L+B+1}^{L+B+K} \|\varepsilon_i - \hat{\varepsilon}_i\|_2^2$;
    Update $\theta$ using gradients of $\mathcal{L}$;

---

# 4 Experiments

## 4.1 Overview

We evaluate our proposed method on two recent large-scale text-to-video generation models: CogVideoX-5B and Wan-14B. Our experiments span a diverse range of conditional generation tasks, including:

- **Image-to-Video (I2V)**: e.g., character-to-video generation, object-to-motion, virtual try-on, ad-video generation.
- **Video-to-Video (V2V)**: e.g., video style transfer, action transfer, toonification.

A key strength of TIC-FT is its ability to operate in the *few-shot regime*. We fine-tune models with as few as 10–30 training samples and fewer than 1,000 training steps—requiring less than one hour of training time for CogVideoX-5B on a single A100 GPU.

We use both real and synthetic datasets for evaluation and demonstrations. Real datasets include SSv2 [30] and manually curated paired videos, while synthetic datasets are created using models such as GPT-4o image generation [31] and Sora [32] (e.g., translating real images into stylized videos). Each task is provided with 20 condition–target pairs. Additional details are provided in the Appendix.

We compare TIC-FT with three representative fine-tuning methods for conditional video generation. While CogVideoX-5B and Wan-14B are among the most recent and powerful text-to-video diffusion models, most existing editing or fine-tuning approaches have not been evaluated on such large-scale backbones. To ensure meaningful comparisons, we reimplement the following representative baselines.

**ControlNet** [7, 9]. We include ControlNet as a baseline because a large number of recent methods are built upon it or extend its core architecture [16, 17, 14]. It is a widely adopted framework that

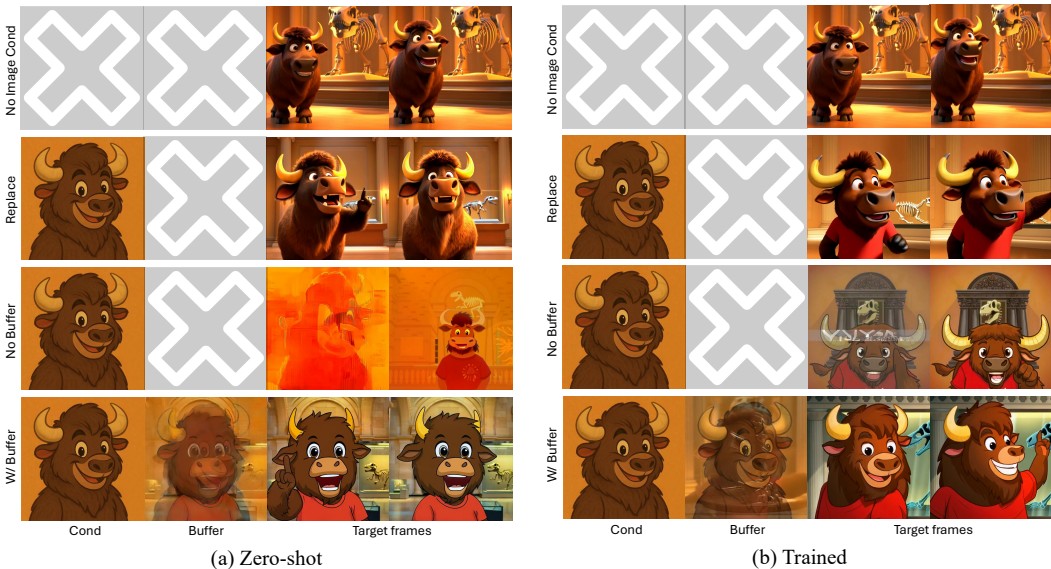

(a) Zero-shot                              (b) Trained

Figure 2: (a) Zero-shot comparison of our method (last row) with variants: without buffer frames and with an SDEdit-style inpainting strategy ("Replace", second row). Buffer frames enable smoother transitions and better condition preservation. (b) Corresponding results after fine-tuning.

introduces an external reference network and zero-convolution layers to inject conditioning signals, enabling the model to preserve fine-grained visual details while integrating external guidance.

**Fun-pose**[10]. A simple yet widely adopted strategy is to concatenate the condition and target latents, as seen in many recent methods[33, 34, 35]. However, this approach requires architectural modifications and extensive retraining, which is infeasible in low-data regimes (e.g., 20 samples). Since training such a model from scratch yields extremely poor results, a direct comparison would be uninformative. Instead, we adopt `Fun-pose`—a variant of CogVideoX and Wan that has already been finetuned to accept reference videos—effectively giving it a significant advantage.

**SIC-FT-Replace**[8, 24]. This method performs spatial in-context fine-tuning by training the model to predict videos arranged as spatial grids. At inference time, the ground-truth condition is noised and repeatedly injected into the condition grid slot at each denoising step, following an SDEdit-style replacement strategy[25], while the remaining grid elements are progressively denoised. This approach represents a recent trend in applying in-context fine-tuning techniques to diffusion models.

## 4.2 Results

We conduct quantitative evaluations using CogVideoX-5B as the base model, focusing on two I2V tasks—object-to-motion and character-to-video—as shown in Table 1. For V2V, we evaluate performance on a style transfer task (real videos to animation), summarized in Table 2. All models are fine-tuned using LoRA (rank 128) with 20 training samples over 6,000 steps, a batch size of 2, and a single NVIDIA H100 80GB GPU. Inference is conducted with 50 denoising steps.

To assess video quality comprehensively, we use three categories of evaluation metrics: VBench [36], GPT-4o [31], and Perceptual similarity scores. VBench provides human-aligned assessments of temporal and spatial coherence, including subject consistency, background stability, and motion smoothness. GPT-4o leverages a multimodal large language model to rate aesthetic quality, structural fidelity, and semantic alignment with the prompt. Perceptual metrics quantify low- and high-level visual similarity between condition and target frames, including CLIP-I and CLIP-T (for image/text alignment), LPIPS and SSIM (for perceptual similarity), and DINO (for structural consistency). However, we omit Perceptual metrics when evaluating tasks like object-to-motion, where different viewpoints may reduce similarity scores despite correct semantics.

Our model achieves strong performance even with limited training, showing competitive results after only 2,000 training steps—unlike other baselines that require significantly more optimization to reach similar quality. Additional comparisons under this low-data, low-compute regime are presented in the Appendix. Despite being conditioned on reference frames, Fun-pose and ControlNet exhibit poor

Table 1: Comparison on VBench, GPT-4o, and perceptual similarity metrics for I2V tasks.

| Method | VBench | | | GPT-4o | | | Perceptual similarity | | | | |
|---|---|---|---|---|---|---|---|---|---|---|---|
| | subject consistency | background consistency | motion smoothness | aesthetic quality | structural similarity | semantic similarity | CLIP-I | CLIP-T | LPIPS↓ | SSIM | DINO |
| ControlNet [7, 9] | 0.9658 | 0.9600 | 0.9926 | 3.87 | 2.69 | 2.69 | 0.7349 | 0.2903 | 0.6535 | 0.3477 | 0.3427 |
| Fun-pose [10] | 0.9508 | 0.9598 | 0.9910 | 4.09 | 2.87 | 3.21 | 0.7714 | 0.3099 | 0.6339 | 0.3575 | 0.3866 |
| SIC-FT-Replace [8, 24] | 0.9513 | 0.9676 | 0.9921 | 4.10 | 2.42 | 2.95 | 0.7993 | 0.3064 | 0.6190 | 0.4455 | 0.4246 |
| TIC-FT-Replace | 0.9580 | 0.9702 | 0.9926 | 4.08 | 2.00 | 2.48 | 0.7925 | 0.3127 | 0.6165 | 0.4123 | 0.4221 |
| TIC-FT (w/o Buffer) | 0.9474 | 0.9686 | 0.9892 | 4.05 | 3.05 | 3.53 | 0.7573 | 0.2986 | 0.6242 | 0.4058 | 0.4160 |
| TIC-FT (2K) | 0.9505 | 0.9696 | 0.9920 | 4.03 | 3.08 | 3.54 | 0.8066 | 0.3135 | 0.6162 | 0.4203 | 0.4240 |
| TIC-FT (6K) | **0.9672** | **0.9729** | **0.9930** | **4.13** | **3.14** | **3.63** | **0.8329** | **0.3143** | **0.4332** | **0.5917** | **0.5530** |

Table 2: Comparison on VBench, GPT-4o, and perceptual similarity metrics for V2V tasks.

| Method | VBench | | | GPT-4o | | | Perceptual similarity | | | | |
|---|---|---|---|---|---|---|---|---|---|---|---|
| | subject consistency | background consistency | motion smoothness | aesthetic quality | structural similarity | semantic similarity | CLIP-I | CLIP-T | LPIPS↓ | SSIM | DINO |
| ControlNet [7, 9] | 0.9553 | 0.9545 | 0.9854 | 3.44 | 2.23 | 2.41 | 0.6221 | 0.2727 | 0.5434 | 0.3494 | 0.2839 |
| Fun-pose [10] | 0.9679 | 0.9675 | 0.9902 | **4.24** | 2.68 | 3.23 | 0.7260 | 0.3018 | 0.5179 | 0.3328 | 0.4369 |
| SIC-FT-Replace [8, 24] | 0.9609 | 0.9655 | 0.9853 | 3.99 | 2.44 | 2.94 | 0.7368 | **0.3198** | 0.5998 | 0.2192 | 0.4025 |
| TIC-FT-Replace | 0.9584 | 0.9696 | 0.9802 | 3.93 | 2.33 | 2.92 | 0.7305 | 0.3015 | 0.6373 | 0.2526 | 0.3673 |
| TIC-FT (w/o Buffer) | 0.9479 | 0.9571 | 0.9744 | 3.81 | 2.66 | 3.20 | 0.7471 | 0.3020 | 0.4687 | 0.3800 | 0.4429 |
| TIC-FT (2K) | 0.9439 | 0.9600 | 0.9865 | 3.85 | 3.67 | 4.37 | 0.8174 | 0.3132 | 0.2970 | 0.5546 | 0.6089 |
| TIC-FT (6k) | **0.9736** | **0.9743** | **0.9935** | 3.99 | **3.90** | **4.41** | **0.8794** | 0.3118 | **0.2251** | **0.6541** | **0.6745** |

condition fidelity. While their outputs appear visually plausible—as indicated by favorable VBench and GPT-4o scores—they consistently underperform in Perceptual similarity metrics, highlighting a lack of alignment with the conditioning input. This is especially problematic for ControlNet, which relies on strict spatial alignment and thus struggles in tasks such as character-to-video and object-to-motion, where viewpoint shifts are common. SIC-FT-Replace[8, 24] also performs suboptimally in I2V settings, as it requires replicating a single frame across a spatial grid—leading to high memory usage and inefficient training. Furthermore, its reliance on SDEdit [25]-style sampling during inference degrades generation quality and weakens condition adherence.

We supplement quantitative results with qualitative comparisons across I2V and V2V tasks in Figure 3. We also present additional scenarios—including virtual try-on, ad-video generation, and action transfer—are illustrated in Figures 1 and 4.

Overall, our proposed **TIC-FT** consistently outperforms prior methods across diverse tasks, with both quantitative metrics and qualitative examples supporting its superior condition alignment and generation quality. More results and task-specific details are provided in the Appendix.

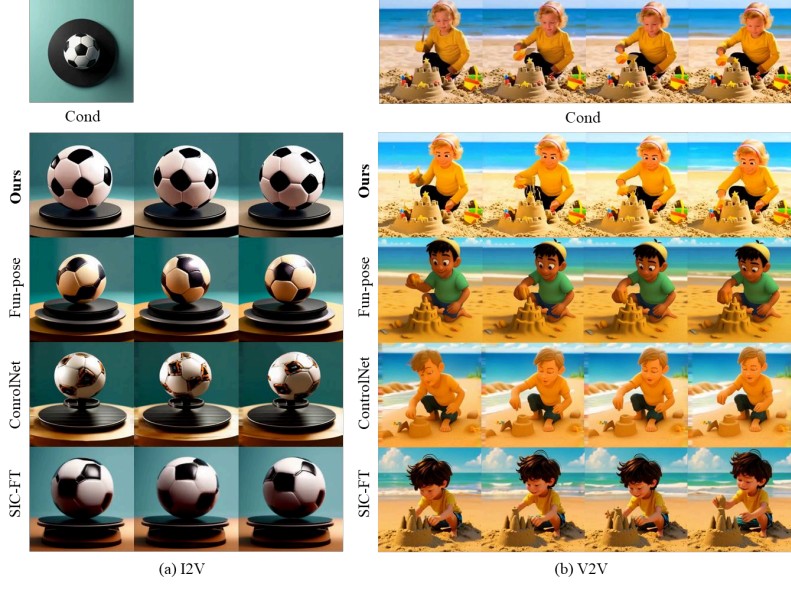

Figure 3: Qualitative comparison between our method and baseline approaches.

Table 3: Quantitative comparison of varying numbers of condition and buffer frames on I2V task.

| #Cond | CLIP-I ↑ | LPIPS ↓ | SSIM ↑ | DINO ↑ | #Buffer | Dynamic Degree ↑ | CLIP-I ↑ | LPIPS ↓ | SSIM ↑ | DINO ↑ |
|---|---|---|---|---|---|---|---|---|---|---|
| 1 | 0.8329 | 0.7493 | 0.5917 | 0.5530 | 1 | 0.72 | 0.7864 | 0.6123 | 0.4128 | 0.4237 |
| 3 | 0.8332 | 0.7390 | 0.5918 | 0.5531 | 3 | 0.73 | 0.7812 | 0.6112 | 0.4130 | 0.4259 |
| 6 | 0.8371 | 0.7360 | 0.6078 | 0.5606 | 6 | 0.77 | 0.7695 | 0.6121 | 0.4127 | 0.4170 |
| 9 | 0.8396 | 0.7346 | 0.6083 | 0.5643 | 9 | 0.82 | 0.7544 | 0.6232 | 0.3952 | 0.4152 |

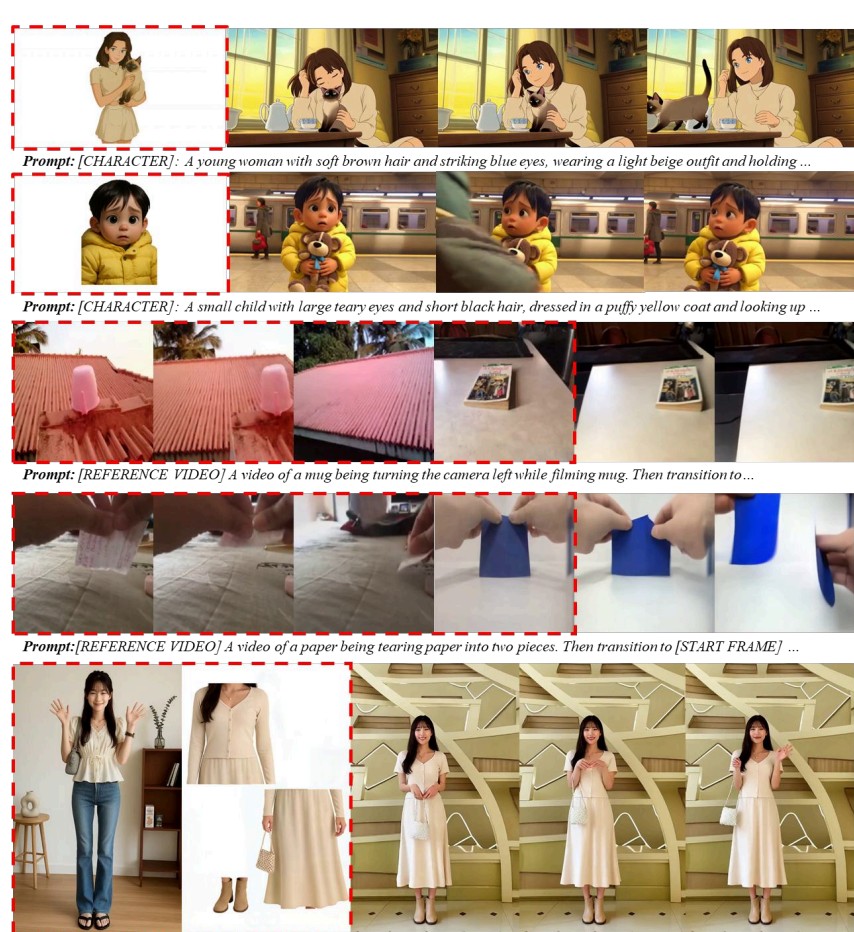

**Prompt:** *[CHARACTER]: A young woman with soft brown hair and striking blue eyes, wearing a light beige outfit and holding ...*

**Prompt:** *[CHARACTER]: A small child with large teary eyes and short black hair, dressed in a puffy yellow coat and looking up ...*

**Prompt:** *[REFERENCE VIDEO] A video of a mug being turning the camera left while filming mug. Then transition to ...*

**Prompt:** *[REFERENCE VIDEO] A video of a paper being tearing paper into two pieces. Then transition to [START FRAME] ...*

**Prompt:** *[IMAGE] A young woman smiles with hands raised near her shoulders, dressed in a short-sleeved white blouse, blue ...*

Figure 4: Demonstration of our method on character-to-video, action transfer, and virtual try-on.

## 4.3 Ablation study

**Zero-Shot Validation of Temporal Concatenation**   We validate the effectiveness of our temporal concatenation design with buffer frames by assessing its zero-shot performance. If the model successfully leverages the pretrained capabilities of video diffusion models, it should generate plausible outputs even without any additional training.

As shown in Figure 2(a), our method with buffer frames (last row) generates target frames that align well with the given condition—demonstrating strong zero-shot performance. In contrast, removing the buffer frames leads to abrupt noise-level discontinuities between condition and target regions, causing the target frames to degrade and the condition information to be poorly preserved. We also compare with zero-shot inpainting methods similar to SDEdit, denoted as "Replace" (second row), which similarly fails to propagate condition signals into the generated frames.

Furthermore, in Figure 2(b), we observe that strong zero-shot performance correlates with better results after fine-tuning. Our method with buffer frames consistently outperforms other variants: models trained without buffer frames begin with blurry target frames, and the "Replace" strategy fails to apply condition information effectively even after training.

**Impact of Condition and Buffer Frames.** We additionally perform quantitative analysis to investigate how the numbers of condition and buffer frames affect performance. As shown in Table 3, increasing condition frames slightly improves condition fidelity, while more buffer frames enhance video dynamics at the cost of condition adherence. Using one condition frame and three buffer frames achieves the best balance between visual quality, motion dynamics, and efficiency.

## 5  Conclusion and Limitation

**Conclusion.** Temporal In-Context Fine-Tuning (TIC-FT) offers an efficient framework for adapting pretrained text-to-video diffusion models to diverse conditional video generation tasks that leverage contextual information from demonstrations. TIC-FT temporally concatenates condition and target frames with intermediate buffer frames to better align with the pretrained model distribution. This design enables a unified and efficient framework for diverse conditional video generation tasks, consistently outperforming existing methods in both condition fidelity and visual quality. TIC-FT achieves these gains without architectural modification and operates with significantly lower computational cost.

**Future Directions.** TIC-FT currently assumes paired condition–target data per task. Generalizing to multi-task or zero-shot settings, where a single TIC-FT model can adapt to heterogeneous tasks without retraining, represents an exciting avenue for future research. Furthermore, while our method has been validated on both synthetic and real-world datasets (e.g., SSv2), expanding datasets to more complex human motion and long-horizon dynamics will further strengthen its generalization ability.

**Limitations.** Unlike In-Context Learning (ICL) in Large Language Models (LLMs), which infers patterns through contextual reasoning without updating model parameters, TIC-FT requires actual fine-tuning during adaptation—similar to In-Context Fine-Tuning approaches used in image generative models [8] . Exploring In-Context Learning (ICL) in video diffusion models could be an interesting future direction.

## Acknowledgments and Disclosure of Funding

This work was supported by Institute for Information & Communications Technology Planning & Evaluation (IITP) grant funded by the Korea government (MSIT) (RS-2019-II190075, Artificial Intelligence Graduate School Program (KAIST)), the National Research Foundation of Korea (NRF) grant funded by the Korea government (MSIT) (No. RS-2025-00555621), and i-Scream Media.

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

# A  Technical Appendices and Supplementary Material

## A.1  Training Details

All models are fine-tuned using an NVIDIA H100 GPU. Our method builds on the CogVideoX-5B backbone and is fine-tuned with LoRA (rank 128), resulting in approximately 130M trainable parameters. Training with 49 frames requires roughly 30GB of GPU memory. For ControlNet, we apply LoRA with the same rank, yielding a comparable parameter count of around 150M, and requiring approximately 60GB of GPU memory. For Fun-pose, we use the official full fine-tuning setup, which consumes around 75GB of GPU memory.

## A.2  Training Amount vs. Performance

This section demonstrates the training efficiency of our method compared to ControlNet. Figure 5 presents performance curves for various metrics—including CLIP-T, CLIP-I, SSIM, DINO, and LPIPS—plotted against training time. Our method consistently outperforms ControlNet across all metrics at equivalent training durations. Moreover, with the exception of CLIP-T, all metrics show a clear upward trend, indicating continued improvement with more training. In contrast, ControlNet exhibits no such trend, suggesting that its training style tends to overfit and struggles to generalize under limited data regimes.

## A.3  Ablation Study

We conduct ablation study on various buffer frame designs. Specifically, we compare our default setting—using a uniformly increasing noise schedule—with alternative strategies: (1) a constant noise level t for all buffer frames (denoted as Constant-$t$, where $T = 100$), and (2) linear-quadratic schedules with concave or convex profiles. Figure 6 presents both zero-shot and fine-tuned results for these configurations. While all variants produce reasonable target frames, we observe that the convex schedule and the constant-25 baseline exhibit poor condition alignment and noticeable artifacts in the zero-shot setting. After fine-tuning, all methods perform comparably, though our default setting with uniformly increasing noise remains preferred. Quantitative results after training are presented in Table 4 and Table 5 for the I2V and V2V tasks, respectively.

We also evaluate the effect of varying the number of buffer frames, ranging from 1 to 5, denoted as Buffer-$n$ in Figure 7. In the zero-shot setting, we observe that all configurations perform comparably overall; however, shorter buffers tend to produce noisier transitions, likely due to abrupt scene changes. Conversely, longer buffers show a tendency to weaken the influence of the condition. After fine-tuning, all variants produce similarly high-quality results.

## A.4  Dataset

For the object-to-motion task, we use the DTU dataset [37] (**License:** Non-commercial research use only). For character-to-video, keyframe interpolation, and ad video generation tasks, we manually collected condition–video pairs tailored to each task. For action transfer, we curate videos from SSv2 [30] (**License:** Research use only, non-commercial). In the video style transfer task, we first synthesize starting frames using FLUX.1-dev [38] (**License:** Non-commercial License), and then generate paired videos using SoRA [32] (**License:** Proprietary; use governed by OpenAI terms of service), and Wan2.1 [6] (**License:** Apache 2.0). Each task is trained on 30 samples. All videos contain 49 frames at 10 frames per second (fps), resized to either 480×480 or 848×480 while preserving the original aspect ratio.

For evaluation and demonstration, we use image and video conditions that are not part of the training set. These include both manually collected images and synthesized ones generated using GPT-4o, FLUX, and Sora. For the action transfer task, we use unseen video samples from SSv2 [30]. Quantitative evaluations are conducted on 100 samples. For image-based metrics such as CLIP and LPIPS, scores are computed on a per-frame basis and then averaged to obtain the final results.

Training and evaluation prompts are generated using GPT-4o. Each prompt is structured to encompass the condition, buffer, and target frames, with condition and buffer frames denoted as `[CONDITION]` and target frames as `[VIDEO]`. Below is the full prompt used for the sample in the ablation study:

**TIC-FT prompt**

*This animated clip demonstrates the transformation of a static character illustration into a lively and expressive animated figure; [CONDITION] the condition image showcases a cheerful cartoon-style young buffalo with thick brown fur, curved yellow horns, and a big, friendly smile. The character's wide eyes and upright posture are set against a warm orange background, giving it a lively and playful presence. [VIDEO] the video animates the buffalo inside a grand museum, where it wears a red t-shirt and points excitedly at a large dinosaur skeleton behind glass. Its eyes are wide with curiosity and its mouth open in awe, while elegant stone columns and soft lighting emphasize the sense of wonder and fascination with history.*

## A.5 Task Descriptions

We detail the construction of data and latent sequences for each conditional video generation task used in our experiments. All tasks are configured with a total of 13 latent frames, corresponding to 49 video frames. While this number can be adjusted based on application needs, we adopt the 13-frame setting throughout for implementation simplicity and consistency. The initial latent sequence comprises condition frames, intermediate buffer frames, and noised target frames. An exception is the action transfer task, where buffer frames are omitted, as the last condition frame serves as the starting frame of the target sequence. The specific configurations for each task are described below.

**Image-to-Video** This task aims to generate a full video conditioned on a single image. The video need not begin directly from the image's visual content; instead, the image may represent a high-level concept such as a character profile or an object viewed from the top, with the video depicting novel dynamics (e.g., a rotating 360° view).

A single reference image is replicated to occupy the first 4 latent frames, followed by 9 target frames.

- Clean condition: 1 frame (from the image)
- Buffer: 3 frames (noised condition)
- Target: 9 frames (pure noise)

We visualize the initial latent frames and their denoising process in Figure 8.

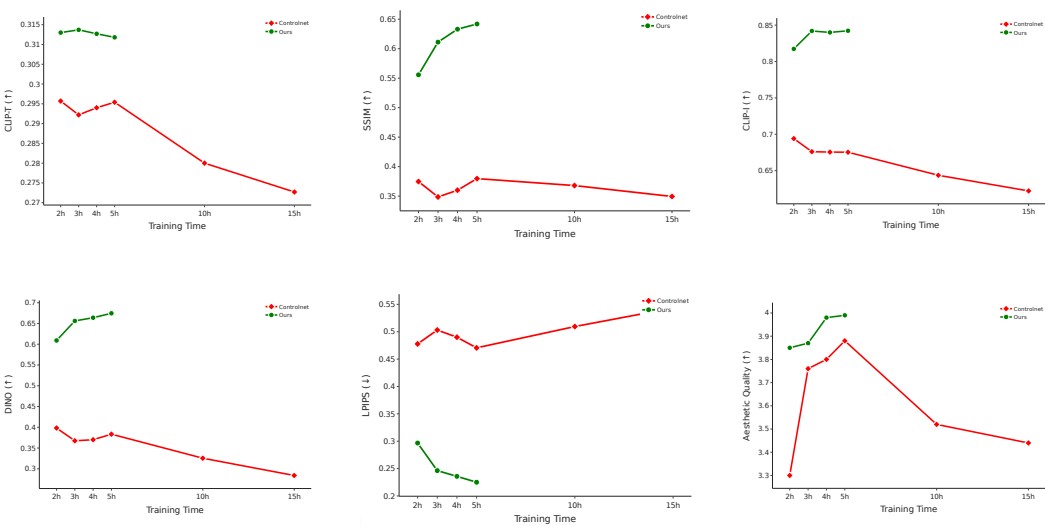

Figure 5: Performance curves for CLIP-T, CLIP-I, SSIM, DINO, and LPIPS metrics plotted against training time. Our method consistently outperforms ControlNet across all metrics at equivalent training durations.

Table 4: Ablation study of constant noise scheduling for buffer frames, evaluated on I2V tasks using VBench, GPT-4o, and perceptual/similarity metrics.

| Method | VBench | | | GPT-4o | | | Perceptual similarity | | | | |
|---|---|---|---|---|---|---|---|---|---|---|---|
| | subject consistency | background consistency | motion smoothness | aesthetic quality | structural similarity | semantic similarity | CLIP-I | CLIP-T | LPIPS↓ | SSIM | DINO |
| Ours | **0.9672** | 0.9729 | **0.9930** | **4.13** | **3.14** | 3.63 | **0.8329** | **0.3143** | **0.4332** | **0.5917** | **0.5530** |
| Constant-25 | 0.9516 | 0.9724 | 0.9920 | 4.09 | 2.81 | 3.45 | 0.7734 | 0.3062 | 0.6088 | 0.4240 | 0.4202 |
| Constant-50 | 0.9509 | **0.9740** | 0.9915 | 4.05 | 3.01 | 3.51 | 0.7760 | 0.3010 | 0.6157 | 0.4188 | 0.4228 |
| Constant-75 | 0.9511 | 0.9722 | 0.9917 | 4.02 | 3.07 | **3.68** | 0.7725 | 0.3003 | 0.6148 | 0.4250 | 0.4259 |

Table 5: Ablation study of constant noise scheduling for buffer frames, evaluated on V2V tasks using VBench, GPT-4o, and perceptual/similarity metrics.

| Method | VBench | | | GPT-4o | | | Perceptual similarity | | | | |
|---|---|---|---|---|---|---|---|---|---|---|---|
| | subject consistency | background consistency | motion smoothness | aesthetic quality | structural similarity | semantic similarity | CLIP-I | CLIP-T | LPIPS↓ | SSIM | DINO |
| Ours | **0.9736** | **0.9743** | **0.9935** | **3.99** | **3.90** | **4.41** | **0.8794** | 0.3080 | **0.2298** | **0.6541** | **0.6596** |
| Constant-25 | 0.9539 | 0.9652 | 0.9873 | 3.90 | 3.55 | 4.20 | 0.8037 | 0.3103 | 0.2744 | 0.5785 | 0.6083 |
| Constant-50 | 0.9524 | 0.9652 | 0.9886 | 3.88 | 3.86 | 4.31 | 0.8460 | **0.3153** | 0.2364 | 0.6039 | 0.6528 |
| Constant-75 | 0.9327 | 0.9552 | 0.9821 | 3.69 | 3.60 | 4.25 | 0.8330 | 0.3142 | 0.2797 | 0.5707 | 0.6368 |

**Video Style Transfer** This video-to-video task transforms the visual style of a source video into that of a target domain (e.g., converting a realistic video into an animated version) while preserving motion and structure.

The first 7 latent frames are taken from a source video and the remaining 6 from a style-transferred version.

- Clean condition: 4 frames (from the source video)
- Buffer: 3 frames (noised condition)
- Target: 6 frames (pure noise)

We visualize the initial latent frames and their denoising process in Figure 9.

**In-Context Action Transfer** This task generates a video that continues a novel scene using motion inferred from a source video. Given a reference action and the first frame of a new environment, the model synthesizes future frames that imitate the observed motion within the new context.

The first 6 latent frames are from a reference action video, the 7th is the first frame of a novel scene, and the rest are the continuation.

- Clean condition: 6 frames (from the reference action video)
- Query frame: 1 clean frame (from the novel scene)
- Target: 6 frames (pure noise)

*No buffer frames are used in this task, as the first frame of the target video is explicitly provided as part of the condition.* We visualize the initial latent frames and their denoising process in Figure 10.

**Keyframe Interpolation** This task fills in intermediate frames between sparse keyframes to produce a temporally coherent video. The goal is to ensure smooth transitions between given keyframes.

Four keyframes are replicated to fill the first 7 latent frames, and the remaining 6 are interpolated.

- Clean condition: 4 frames (replicated keyframes)
- Buffer: 3 frames (noised condition)
- Target: 6 frames (pure noise)

We visualize the initial latent frames and their denoising process in Figure 11.

**Multiple Image Conditions** This task takes two distinct types of image conditions—such as a person and clothing, or a person and an object—and generates a target video that reflects the combination of

both. This setup is useful for applications like virtual try-on (VITON) or ad video synthesis, where two semantic entities must be jointly represented in motion.

The first 3 latent frames are derived from the first condition image, and the next 4 from the second condition image.

- Clean condition: 4 frames (3 from the first image, 1 from the second)
- Buffer: 3 frames (noised condition)
- Target: 6 frames (pure noise)

*Note that the number of condition sources is not limited to two; the framework supports arbitrary multi-condition setups.* We visualize the initial latent frames and their denoising process in Figure 12.

## A.6 Broader Impacts and Misuse Discussion

Our TIC-FT method enables efficient adaptation of video diffusion models with minimal data. However, this ease of fine-tuning also introduces risks, particularly the potential misuse for creating deepfakes or misleading synthetic media. Clear usage policies and responsible deployment practices are essential to mitigate societal risks.

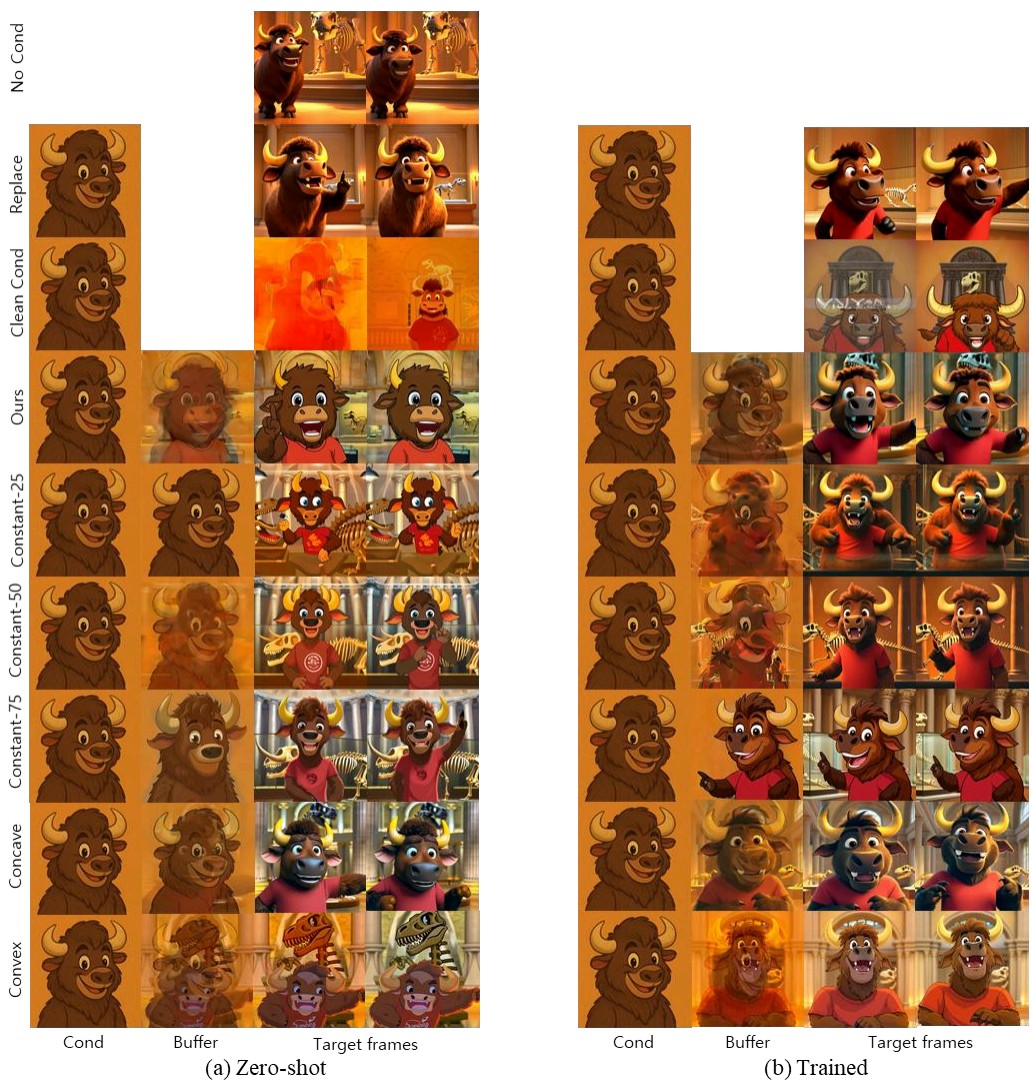

(a) Zero-shot (b) Trained

Figure 6: Qualitative comparison of buffer frame designs in zero-shot and fine-tuned settings.

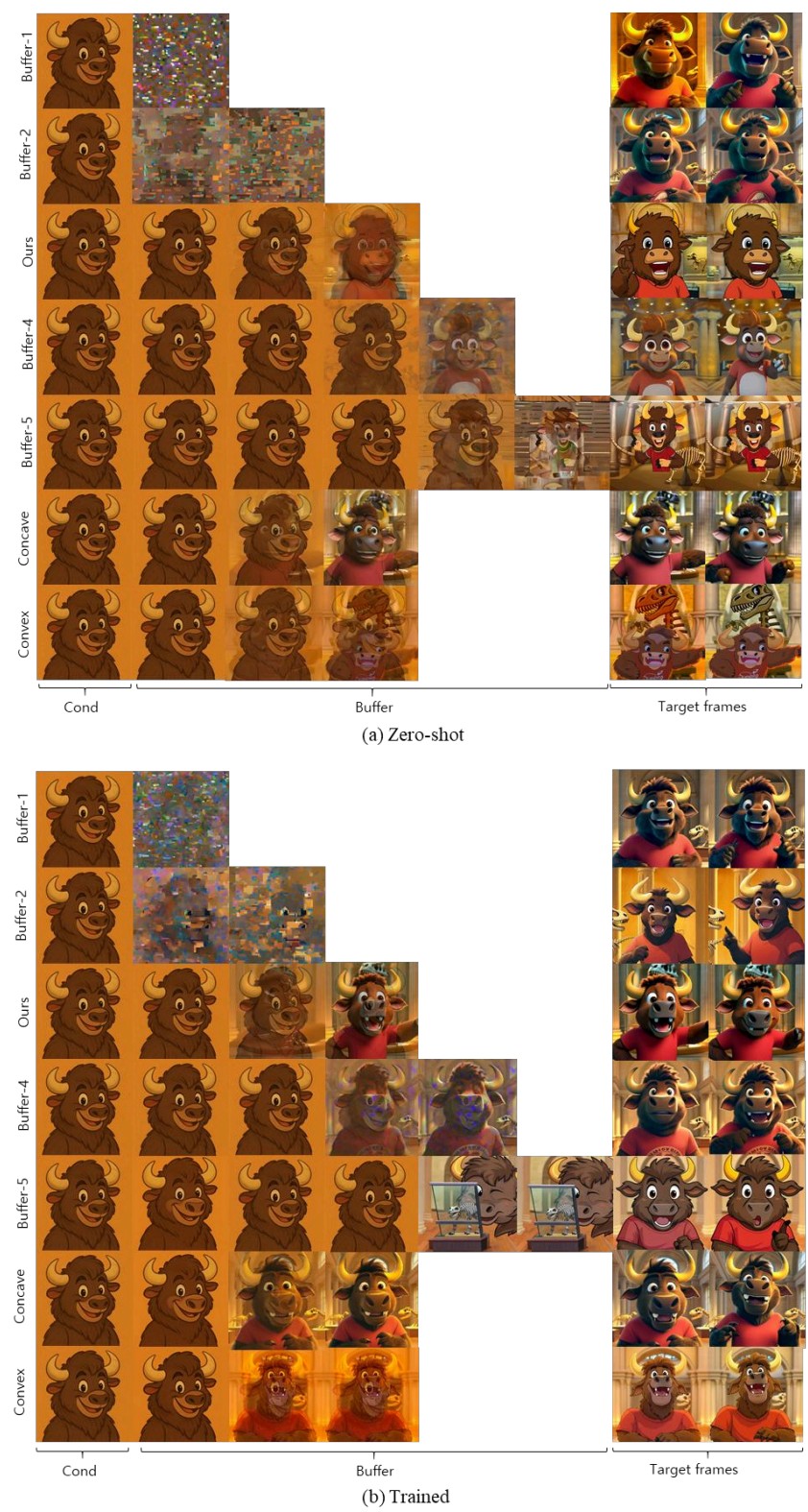

Figure 7: Qualitative comparison of buffer frame designs in zero-shot and fine-tuned settings.

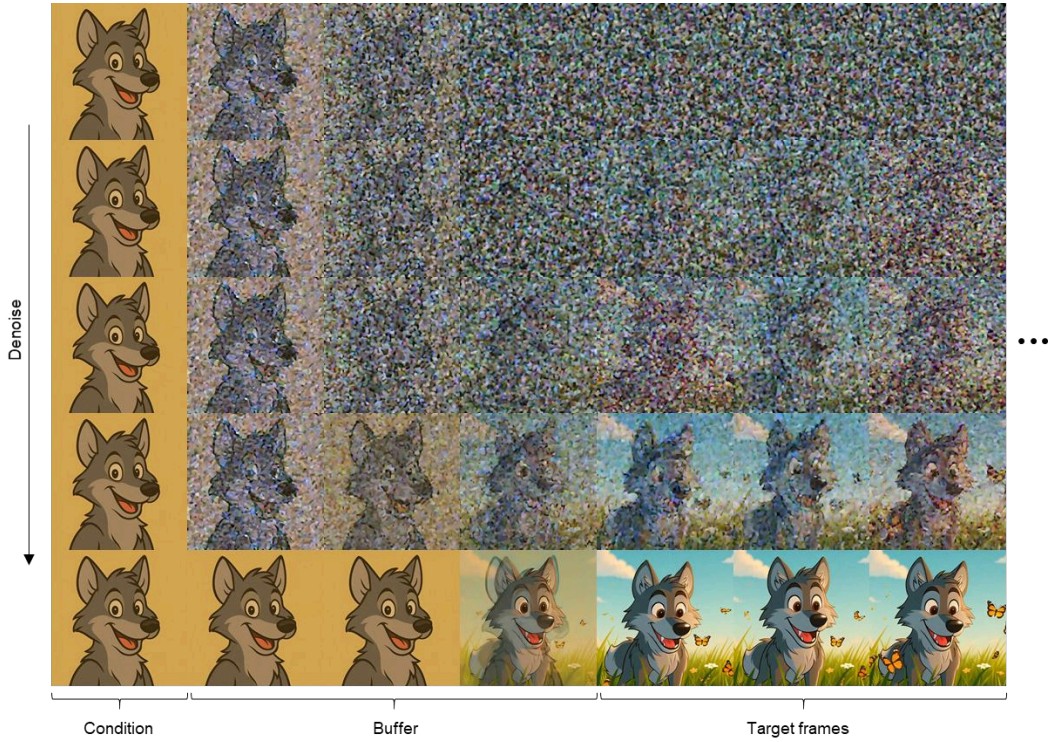

Figure 8: Visual results for initial frames and their denoising process on image-to-video generation. **Prompt:** *[Character] A clear, high-resolution front-facing close-up of a cheerful cartoon-style wolf character, centered against ...*

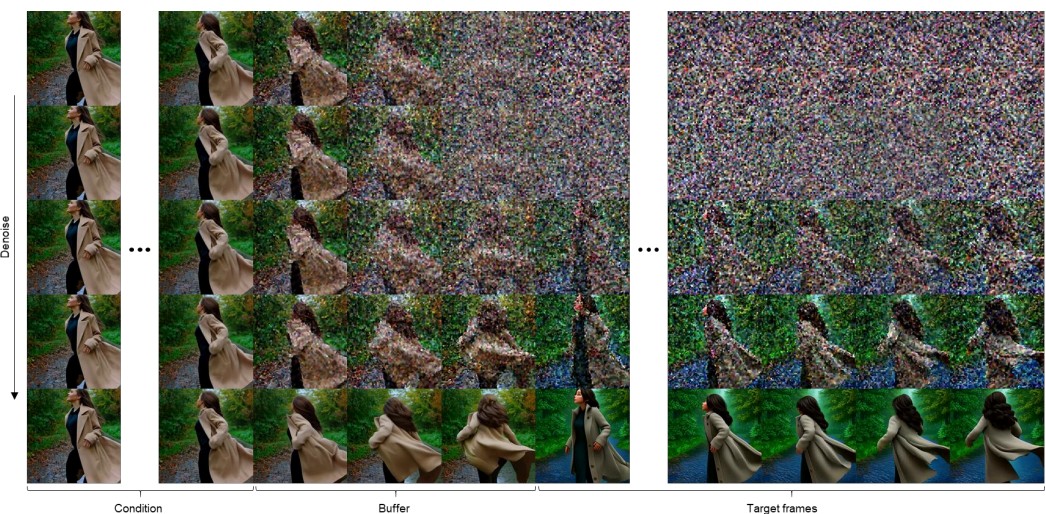

Figure 9: Visual results for initial frames and their denoising process on video style transfer task. **Prompt:** *[VIDEO1] A woman in a tan cloak walks gracefully along a forest path. Her hair flows gently with her movement, and the ...*

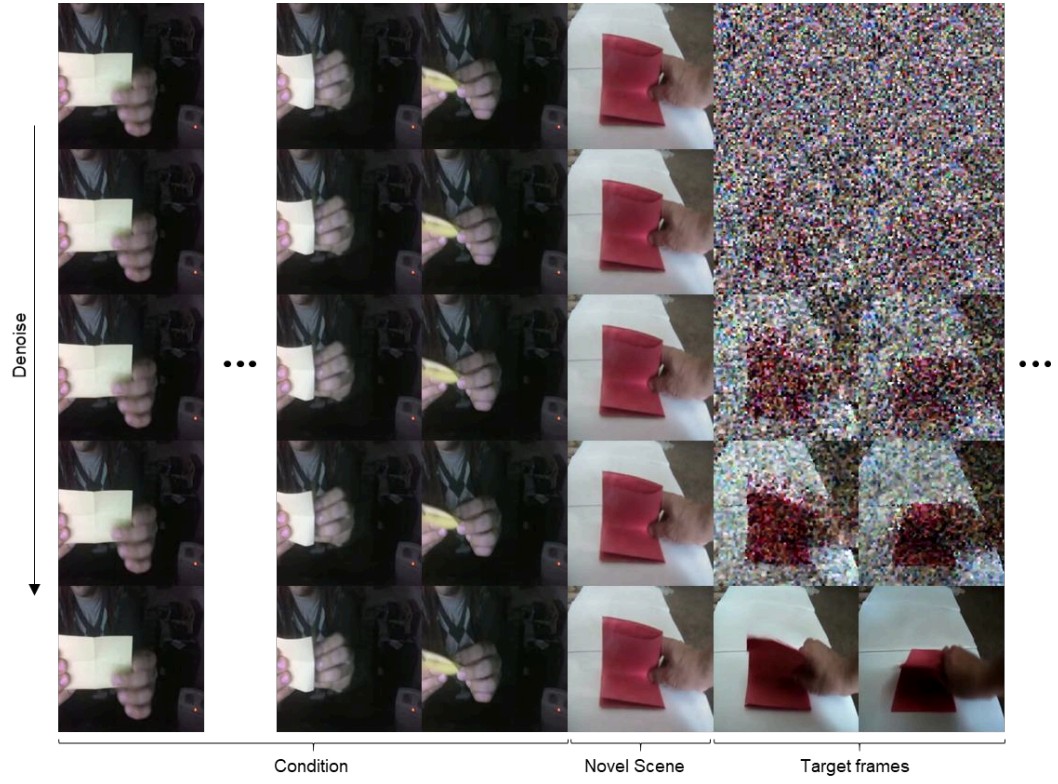

Figure 10: Visual results for initial frames and their denoising process on in-context action transfer task. **Prompt:** *[REFERENCE VIDEO] A white paper is folded in half by a person wearing black sleeves in a dark indoor environment. ...*

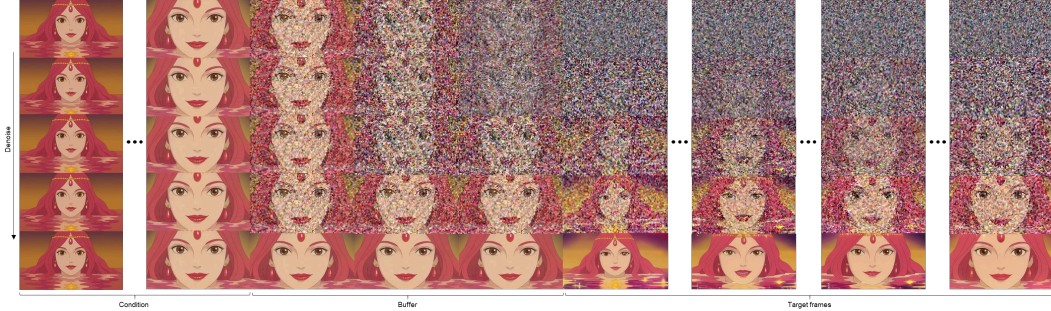

Figure 11: Visual results for initial frames and their denoising process on keyframe interpolation task. **Prompt:** *[VIDEO1] A cartoon woman with red hair and a jeweled headpiece slowly tilts her head and changes facial expressions ...*

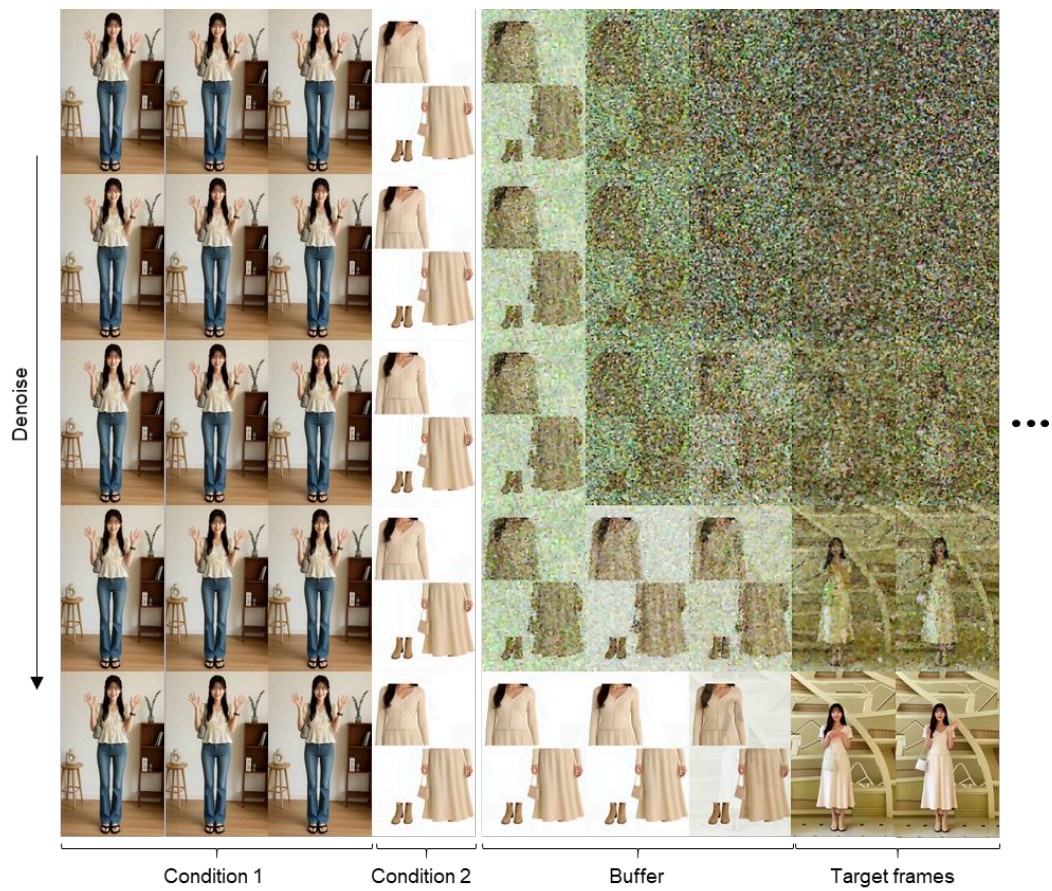

Figure 12: Visual results for initial frames and their denoising process on virtual try-on task. **Prompt:** *[IMAGE] A young woman with long black hair, wearing a cream blouse, blue jeans, and black sandals, smiles with both ...*

