# OpenReview forum: "Temporal In‑Context Fine‑Tuning with Temporal Reasoning for Versatile Control of Video Diffusion Models"
_NeurIPS.cc/2025/Conference — NeurIPS 2025 poster_

### Official Review · Reviewer_juYd · 2025-07-02

**Clarity:** 3
**Significance:** 3
**Originality:** 3
**Rating:** 5
**Confidence:** 3

**Summary:**

This paper presents a novel paradigm for controllable video generation. Similar to in-context learning, this method concatenates conditionaframes to target frames and encourages the model to learn the in-context pattern in conditional frames. This method also introduces a buffer frame between target and condition frames to preserve the pretrained model's capability.

**Questions:**

See weakness.

**Ethical Concerns:**

["NO or VERY MINOR ethics concerns only"]

**Final Justification:**

The authors have addressed my concern in positional encoding, scaling and precise control. This paper is good to accept and I will keep my scores.

**Limitations:**

Yes

**Paper Formatting Concerns:**

No formatting concerns.

**Quality:**

3

**Strengths And Weaknesses:**

Stength:

1) The proposed method is novel and highly extensible. Compared to previous approaches such as ControlNet, it enables multiple applications through simple concatenation without introducing additional parameters.

2) The paper's writing is clear. The method is straightforward and easy to follow.

3) The experiments are sufficient and the performance is good.

Weakness:

(1) Positional Encoding. After concatenation of conditional , buffer and target frames, the total number of frames likely exceeds the original video length supported by the pretrained model. If the video length during fine-tuning differs from that of the pretrained model, it is unclear whether this would cause issues. It would be helpful to clarify how the positional encoding is handled for the longer frames.

(2) Scaling. In this paper, each application requires a set of training data for fine-tuning. Is it possible to jointly fine-tuning on multiple tasks and  achieve zero-shot performance at inference time?

(3) Precise Control. If fine-grained control over each frame (e.g., conditioning each frame on a depth map) is needed, is it feasible under the proposed in-context learning framework?

---

> ### Author Rebuttal · Authors · 2025-07-30
>
> We sincerely thank the reviewer for the encouraging and thoughtful feedback. We appreciate your recognition of the clarity, extensibility, and effectiveness of our method. Below, we address your questions regarding positional encoding for longer sequences, the potential for multi-task joint fine-tuning, and the feasibility of fine-grained frame-level control under our in-context learning framework.
> ***
> > (1) Positional Encoding. After concatenation of conditional , buffer and target frames, the total number of frames likely exceeds the original video length supported by the pretrained model. If the video length during fine-tuning differs from that of the pretrained model, it is unclear whether this would cause issues. It would be helpful to clarify how the positional encoding is handled for the longer frames.
>
> We thank the reviewer for this important technical question.
> This is generally not a concern because recent pretrained video models are trained on videos of diverse lengths and can generate longer sequences using efficient attention mechanisms and/or autoregressive approaches. Our method adds only a few buffer and condition frames, resulting in relatively small computational overhead.
>
> Furthermore, pretrained video models leverage Rotary Positional Encoding (RoPE), which effectively generalizes to sequence lengths beyond those seen during training. Since RoPE relies on relative rather than absolute positional information, it handles extended contexts without significant performance degradation.
>
> Importantly, within the scope of our experiments, the total concatenated sequence length (condition + buffer + target frames) remained well within the normal sequence lengths supported by the pretrained backbones, so we did not encounter any issues. Nevertheless, as long video generation remains a challenging problem with rapidly evolving research, we believe there are opportunities for further exploration in conditioning longer video generation.
>
> ***
> > (2) Scaling. In this paper, each application requires a set of training data for fine-tuning. Is it possible to jointly fine-tuning on multiple tasks and achieve zero-shot performance at inference time?
>
> While not directly within the scope of our current research—which focuses on efficient fine-tuning with minimal data and computation—we believe this is definitely feasible.
> In the image domain, the Kontext [1] model demonstrates how in-context fine-tuning can build a general image editing model capable of zero-shot performance.
>
> Our temporal in-context fine-tuning method works effectively because it leverages the pretrained distribution of video models well, suggesting it could quickly adapt to diverse samples when trained on large-scale diverse datasets.
> One key difference would be that while our current approach requires paired videos (source/condition and target), scaling to multi-task zero-shot performance would likely require triplets of source, target, and text instructions to incorporate diversity and enable controllable generation.
> This represents an exciting direction for future research.
>
> [1] Batifol, Stephen, et al. "FLUX. 1 Kontext: Flow Matching for In-Context Image Generation and Editing in Latent Space." arXiv e-prints (2025): arXiv-2506.
> ***
> > (3) Precise Control. If fine-grained control over each frame (e.g., conditioning each frame on a depth map) is needed, is it feasible under the proposed in-context learning framework?
>
> We appreciate the reviewer's question about the scope of controllability.
> The TIC-FT framework is conceptually flexible enough to support such scenarios and would fall under the video-to-video (V2V) category presented in our paper. For example, one could structure the input as temporally arranged sequences of `[condition frames (depth maps), target frames]` or `[condition frames (depth maps), buffer frames, target frames]`.
>
> However, we should clarify that TIC-FT's primary design goal is not for tasks requiring dense, pixel-aligned conditional inputs like per-frame depth control. For such applications, more direct approaches—such as channel-wise concatenation of conditions or ControlNet-style methods where pixel-aligned condition features are directly added to latent frames—would likely provide more straightforward learning pathways for the model.
>
> ***
> Once again, we are grateful for your valuable feedback. Your questions have prompted us to clarify important aspects of our design and explore directions for future extension. We hope our detailed responses have addressed your concerns and look forward to further discussions.

---

> > ### Comment · Reviewer_juYd · 2025-08-05
> >
> > Thank you for your responses and they have addressed my concerns. This paper is good to accept and I will keep my scores.

---

### Official Review · Reviewer_9QKt · 2025-07-02

**Clarity:** 3
**Significance:** 2
**Originality:** 3
**Rating:** 4
**Confidence:** 3

**Summary:**

This paper introduces Temporal In-Context Fine-Tuning (TIC-FT), a parameter-efficient method for adapting pretrained video diffusion models to diverse conditional generation tasks. Key innovations include:

1. Temporal Context Construction:
   - Concatenates condition frames (e.g., input images/videos) and target frames along the temporal axis
   - Inserts noisy buffer frames with progressive noise levels between condition-target pairs to align with pretrained diffusion dynamics
2. Training-Free Adaptation:
   - Requires no architectural changes or external encoders
   - Achieves strong performance with 10–30 training samples
3. Versatile Control:
   - Validated on image-to-video and video-to-video tasks using billion-scale models (CogVideoX-5B, Wan-14B)
   - Maintains high condition fidelity and visual quality while reducing training/inference costs

TIC-FT establishes temporal concatenation as an effective paradigm for lightweight control of video diffusion models.

**Questions:**

1. Baseline Relevance
   - Q: Why omit video-specific control methods (e.g., ControlVideo, VidControl)? Does TIC-FT outperform them under identical compute?
2. Temporal Artifact Analysis
   - Q: Buffer frames reduce distribution shift but may cause motion flicker. Did you quantify temporal instability (e.g., FLD score)?
3. Long-Sequence Scalability
   - Q: The 10-second limit stems from memory constraints. Could sequence chunking or gradient checkpointing mitigate this?
4. Real-World Generalization
   - Q: Synthetic datasets (GPT-4o/Sora) may bias results. Does performance hold on complex real-world motions?

**Ethical Concerns:**

["NO or VERY MINOR ethics concerns only"]

**Limitations:**

YES.

**Paper Formatting Concerns:**

NO.

**Quality:**

3

**Strengths And Weaknesses:**

Strengths:

- Quality:
  Rigorous validation on billion-scale models across 6+ tasks (style transfer/motion transfer/etc.). Few-shot efficiency is empirically verified.
- Originality:
  Novel temporal concatenation + noisy buffer frames mechanism elegantly aligns pretrained/fine-tuning distributions. Unifies diverse conditioning tasks without architectural changes.
- Significance:
  Democratizes control of large video models for low-resource users. Memory-efficient design.
- Clarity:
  Exceptional methodological description; clean task taxonomy.

Weaknesses:

- Quality:
  - Baseline mismatch: ControlNet (image-centric) and Fun-pose (pre-finetuned) are suboptimal comparators. Missing video-specific adapters (e.g., TADA, VidControl).
  - Limited temporal metrics: Relies on FVD/CLIP – neglects *motion consistency* (e.g., optical flow variance).
- Significance:
  - 10-second input limit restricts long-video applications.

---

> ### Author Rebuttal · Authors · 2025-07-30
>
> We thank the reviewer for the thorough and encouraging evaluation of our work. We appreciate your recognition of our method’s originality, clarity, and practical significance, as well as your valuable suggestions regarding baselines, temporal metrics, and long-video applicability. We address the reviewer’s concerns point by point, specifically focusing on (1) baseline selection, (2) the absence of fine-grained motion metrics, and (3) the restriction related to input sequence length.
> ***
> > Q: Why omit video-specific control methods (e.g., ControlVideo, VidControl)? Does TIC-FT outperform them under identical compute?
>
> We want to clarify that ControlNet-based approached mentioned in the paper, **is the same method as ControlVideo**, and our method does outperform the baseline as presented in the paper, under less compute.
> (The ControlNet we used in our experiments is indeed a video-centric version adapted for a Video DiT architecture, functionally identical to the ControlVideo setup you suggested.)
>
> Current leading methods for conditional video generation primarily follow two main paradigms. The first is the ControlNet-based approach, where a parallel (and sometimes trainable) encoder branch is added to the main model to inject spatial or temporal conditions. The second is the concatenation-based approach, where conditioning information, such as a reference frame's latent representation, is directly concatenated with the input noise tensor during the diffusion process and often requires finetuning the whole model.
>
> To ensure our comparison was representative of the current state-of-the-art, we selected baselines that reflect these two dominant strategies. ControlNet (ControlVideo) was chosen as a representative example of the first paradigm, while Fun-pose, a popular and efficient variant of the second category, was included as a strong representative.
>
> We acknowledge that additional methods exist, but we did not identify any work that significantly deviates from these two paradigms or demonstrates substantially superior performance compared to these established approaches. So we believe this selection provides a fair and robust comparison against the primary methods currently employed in the field. However, we are open to discussion and would be happy to include additional baselines if you have specific suggestions!
>
> (We were unable to find TADA model you mentioned despite an extensive search. It would be grateful if you could provide a reference, and we would be happy to perform a comparison.)
>
> You can also check out GPU Memory Usage Comparison:
>
> | Method | Training VRAM | Inference VRAM |
> | --- | --- | --- |
> | **Ours (TIC-FT)** | **30 GB** | **25 GB** |
> | ControlNet | 60 GB | 43 GB |
> | Fun-pose | 75 GB | 28 GB |
>
> As the table demonstrates, our TIC-FT method operates at a significantly lower computational cost for both training and inference. Despite this high efficiency, TIC-FT achieves superior performance compared to these baselines, as detailed in Table 1 and Table 2 of our main paper.
> ***
> > Q: Buffer frames reduce distribution shift but may cause motion flicker. Did you quantify temporal instability (e.g., FLD score)?
>
> We appreciate the reviewer's focus on detailed temporal evaluation. As a quantitative measure, our main paper's Table 1 and Table 2 already include several VBench metrics that evaluate temporal qualities, such as `subject consistency`, `background consistency`, and `motion smoothness`.
>
> In addition to these metrics, we believe the qualitative results provide the most direct evidence of temporal stability. We kindly ask the reviewer to refer to our supplementary video examples. These videos demonstrate that our method produces temporally clean and smooth results, without introducing the flicker or instability the reviewer was concerned about.
>
> To more directly address the concern about motion and stability, we have conducted further evaluations using the new Video-Bench benchmark[1] (accepted at CVPR 2025), which provides human-aligned scores for `motion effects` and `temporal consistency`.
>
> The results on Video-Bench are presented below:
>
> **Table: Evaluation on Video-Bench**
>
> | method | aesthetic quality | imaging quality | motion effects | temporal consistency |
> | --- | --- | --- | --- | --- |
> | ControlNet | 3.60 | 3.58 | 2.70 | 4.75 |
> | Fun-pose | 3.94 | 4.42 | 3.05 | 4.82 |
> | TIC-FT-Replace | 3.92 | 4.29 | 3.07 | 4.80 |
> | TIC-FT (w/o Buffer) | 3.93 | 4.15 | 3.06 | 4.82 |
> | **TIC-FT** | **3.96** | **4.52** | **3.09** | **4.83** |
>
> Results demonstrate the benefit of our design, and `temporal consistency` shows our method is robust from temporal artifacts.
>
> [1] H. Han et al., “Video‑Bench: Human‑Aligned Video Generation Benchmark,” in Proceedings of the IEEE/CVF Conference on Computer Vision and Pattern Recognition (CVPR), 2025.
> ***
> > Q: The 10-second limit stems from memory constraints. Could sequence chunking or gradient checkpointing mitigate this?
>
> We believe the limitation regarding long video generation is not specific to our method, but rather a common challenge and ongoing research area for all video generation models.
> Recent works have focused on longer video generation by employing more efficient attention mechanisms or autoregressive diffusion approaches. Gradient chunking, as you mentioned, would indeed help since the primary bottleneck is memory usage during chunk generation.
> Improving conditioning for longer videos represents an interesting direction for future research.
>
> ***
> > Q: Synthetic datasets (GPT-4o/Sora) may bias results. Does performance hold on complex real-world motions?
>
> We appreciate the reviewer's concern for evaluation on real-world data. To clarify, **our evaluation was not limited to synthetic data**. For the action transfer task, we curated videos directly from the SSv2 (Something-Something V2) dataset, which is composed entirely of real-world videos of humans performing various actions.
>
> Our method demonstrated strong performance on this real-world dataset. The qualitative results for this task, showcasing TIC-FT's effectiveness on complex, real-world motion, can be found in Figure 10 in the Appendix.
> If you are aware of any other real-world experiments that would enhance our evaluation, we would welcome your suggestions!
> ***
> Once again, we are grateful for your thoughtful and balanced feedback. Your comments have helped us further strengthen our work,  and we remain open to additional experiments and further discussion.

---

### Official Review · Reviewer_ngnG · 2025-07-03

**Clarity:** 3
**Significance:** 3
**Originality:** 3
**Rating:** 5
**Confidence:** 4

**Summary:**

- Many existing methods for conditional video generation rely on large training datasets and introduce additional architectural modifications. To address these limitations, this paper proposes Temporal In-Context Fine-Tuning (TIC-FT), an efficient and versatile approach for adapting pretrained video diffusion models to diverse conditional generation tasks.
- Extensive experiments show that TIC-FT outperforms existing baselines in both condition fidelity and visual quality.

**Questions:**

Have the pretrained video diffusion backbones been trained on any conditional generation tasks (i.e., sth like the downstream tasks of this paper) except for general video generation? If the former, how does this influence the effectiveness of TIC-FT?

**Ethical Concerns:**

["NO or VERY MINOR ethics concerns only"]

**Final Justification:**

I have read the reviews from others as well as the authors' responses, which addressed most of my concerns. I think the paper is good to be accepted to NeurIPS.

**Limitations:**

yes

**Quality:**

3

**Strengths And Weaknesses:**

**Strengths**
- The paper introduces a simple yet effective method for adapting pretrained video diffusion models to diverse conditional generation tasks.
- TIC-FT shows strong empirical performance with small training data, offering a highly efficient fine-tuning strategy.
- The proposed method avoids modifying the model architecture, making it easy for adoption.

**Weaknesses**
- L168–169 describe the inference procedure as “iteratively identifying the frames currently at the maximal noise level t and applying the video diffusion sampler exclusively to those frames”. It seems like the buffer frames need to be denoised iteratively, possibly one by one, because they have different noise levels? This seems to increase the inference time complexity? What's the inference time complexity of TIC-FT, and how does it compare to baseline methods?
- It is unclear to me how the number of buffer frames is determined during training and inference.
- It is unclear whether, in cases with multiple conditional frames, does the model only use the first conditional frame to construct the buffer frames? if so, how does the model use multiple conditional frames when constructing the buffer frames?

---

> ### Author Rebuttal · Authors · 2025-07-30
>
> We sincerely thank the reviewer for the thoughtful evaluation and for highlighting both the strengths and the potential areas for clarification in our work. We address each of the concerns in detail and provide clarifications regarding the inference procedure, buffer frame configuration, and the handling of multiple conditional frames below.
> ***
> > L168–169 describe the inference procedure as “iteratively identifying the frames currently at the maximal noise level t and applying the video diffusion sampler exclusively to those frames”. It seems like the buffer frames need to be denoised iteratively, possibly one by one, because they have different noise levels? This seems to increase the inference time complexity? What's the inference time complexity of TIC-FT, and how does it compare to baseline methods?
>
> We would like to clarify that our TIC-FT denoising process remains **fully parallel across all frames** and does not introduce additional iterative denoising steps, thereby incurring no significant overhead. To provide a clearer picture of the denoising procedure, we offer the following step-by-step visualization.
>
>
> - **C:** Condition Frame (a clean frame at `t=0`)
> - **B1, B2, B3:** Buffer Frames with initial target noise levels (e.g., `t=25, t=50, t=75`)
> - **T:** Target Frames to be generated (starting at max noise, e.g., `t=100`)
>
> ---
>
> ### **Denoising Process of TIC-FT (Ours)**
>
> This process follows the formal rules described in **Algorithm 1** and **Equation 7** of the paper.
>
> **Frame Structure:** `[ C | B1 | B2 | B3 | T | T | ... ]`
>
> ### `t_global = 100` (Start)
>
> - **Frame Timesteps:** `[0, 25, 50, 75, 100, 100, ...]`
> - **Frames Being Denoised:** **Target Frames (T)**
>
> ### `t_global = 99`
>
> - **Frame Timesteps:** `[0, 25, 50, 75, 99, 99, ...]`
> - **Frames Being Denoised:** **Target Frames (T)**
>
> ... (This continues until `t_global` = 76) ...
>
> ### `t_global = 75`
>
> - **Frame Timesteps:** `[0, 25, 50, 75, 75, 75, ...]`
> - **Frames Being Denoised:** **B3** + **Target Frames (T)**
>
> ### `t_global = 74`
>
> - **Frame Timesteps:** `[0, 25, 50, 74, 74, 74, ...]`
> - **Frames Being Denoised:** **B3** + **Target Frames (T)**
>
> ... (This continues until `t_global` = 51) ...
>
> ### `t_global = 50`
>
> - **Frame Timesteps:** `[0, 25, 50, 50, 50, 50, ...]`
> - **Frames Being Denoised:** **B2** + **B3** + **Target Frames (T)**
>
> ### `t_global = 49`
>
> - **Frame Timesteps:** `[0, 25, 49, 49, 49, 49, ...]`
> - **Frames Being Denoised:** **B2** + **B3** + **Target Frames (T)**
>
> ... (This continues until `t_global` = 26) ...
>
> ### `t_global = 25`
>
> - **Frame Timesteps:** `[0, 25, 25, 25, 25, 25, ...]`
> - **Frames Being Denoised:** **B1** + **B2** + **B3** + **Target Frames (T)**
>
> ### `t_global = 24`
>
> - **Frame Timesteps:** `[0, 24, 24, 24, 24, 24, ...]`
> - **Frames Being Denoised:** **B1** + **B2** + **B3** + **Target Frames (T)**
>
> ... (This process continues down to `t_global` = 1) ...
>
> ### `t_global = 0` (End)
>
> - **Frame Timesteps:** `[0, 0, 0, 0, 0, 0, ...]`
> - **Result:** Final denoised sequence.
>
> ---
>
> As the visualizations illustrate, our method **requires no additional denoising steps**.  The only extra computation comes from additional condition and buffer frames, which is negligible when generating longer videos. For further clarity, visual examples of denoising are provided in Appendix Figures 8–12.
>
> ***
> > It is unclear to me how the number of buffer frames is determined during training and inference.
>
> The choice of using three (latent) buffer frames as the default in our main experiments was based on a careful balance between generation quality and computational overhead, informed by the ablation study presented in our appendix (Figure 7).
>
> Our ablation study on buffer frame count revealed the following trade-offs:
>
> Too few buffer frames (e.g., one): Results in noisier transitions due to abrupt changes between condition and target frames, causing artifacts and degraded quality in target frames.
>
> Too many buffer frames: Unnecessarily increases sequence length, leading to higher computational and memory costs, without further gain in generation quality. Additionally, excessive temporal gaps between target frames may reduce condition adherence.
>
> We also observed that increasing the number of buffer frames enhances video dynamics in I2V tasks. This occurs because additional buffer frames create a temporal gap between the static input and target videos, leading to disentanglement between the two.
>
> | # Buffer | Dynamic Degree |
> | --- | --- |
> | 3 | 0.73 |
> | 6 | 0.77 |
> | 9 | **0.82** |
>
> Intuitively, as the model further disentangles the target frames from the condition frame, the influence of the condition on the generated target video tends to diminish:
>
> | # Buffer | CLIP-I   | LPIPS  | SSIM   | DINO   |
> |----------|--------|--------|--------|--------|
> | 3        |**0.7812** | **0.6112** | **0.4130** | **0.4259** |
> | 6        |0.7695 | 0.6121 | 0.4127 | 0.4170 |
> | 9        | 0.7544 | 0.6232 | 0.3952 | 0.4152 |
>
> In conclusion, more buffer frames lead to disentanglement between condition and target frames. In I2V generation, since condition frames are static, additional buffer frames increase the dynamics of the generated video, while at the cost of reduced condition reflection.
> We determined that 3 buffer frames provide the optimal balance between memory usage, condition reflection, and dynamic degree.
>
> ***
> > It is unclear whether, in cases with multiple conditional frames, does the model only use the first conditional frame to construct the buffer frames? if so, how does the model use multiple conditional frames when constructing the buffer frames?
>
> When multiple conditions are used, the buffer frames are constructed using only the last conditional frame. This design ensures a smooth transition from the condition sequence to the target frames.
> This process is illustrated in Figure 12 of the appendix.
>
>
>
> ***
> Once again, we appreciate the reviewer’s constructive feedback and critical observations. Your feedback will be used to help us refine our explanations. We look forward to further discussions.

---

> > ### Comment · Reviewer_ngnG · 2025-08-02
> >
> > Thank you for the detailed responses. It has addressed most of my concerns. I think the paper is good to be accepted.

---

### Official Review · Reviewer_QrUT · 2025-07-04

**Clarity:** 3
**Significance:** 3
**Originality:** 3
**Rating:** 5
**Confidence:** 4

**Summary:**

The paper proposes a method, called temporal in-context learning (TIC), for conditional video generation, including image-to-video and video-to-video tasks. Essentially, the visual conditions are concatenated with the noisy latents along the time dimension, allowing the generated frames to be viewed as a natural extension of the conditional frame(s). Several buffer frames are introduced between them to smooth out the transition from input conditions to generated frames. This design spares the need to modify the model architecture and can be trained efficiently with just a few samples. Experimental results compared to a few baselines address the effectiveness of the method in consistency and transferability.

**Questions:**

See weaknesses

**Ethical Concerns:**

["NO or VERY MINOR ethics concerns only"]

**Final Justification:**

The authors have addressed most of my concerns. The paper seems interesting and should be accepted from my perspective. Please incorporate the rebuttal content into the next version accordingly.

**Limitations:**

See weaknesses

**Quality:**

2

**Strengths And Weaknesses:**

(+) TIC-FT is a clever approach to enabling image/video conditions for video generation without requiring the introduction of a large number of new parameters. Viewing the generation as a continuation of the visual condition in the temporal dimension provides a new perspective and more flexible design for future work in this domain.

(+) The design of buffer frames also makes sense in providing space for a transition process between conditions and generated frames. The noise schedule is carefully considered to make the transition smoother.

(+) TIC-FT seems like an efficient fine-tuning strategy that works with just tens of training samples in limited GPU hours.

(+) Experimental results demonstrate superiority in consistency and guidance following compared to ControlNet and Fun-pose.


Some concerns about experiments:

(-) Small-scale evaluation set. Each task is evaluated on only 20 condition-target pairs, which may not be very representative of showing the whole picture of the performance superiority. Are there larger scale benchmarks or more standard ones that can be used for evaluation?

(-) Metrics: One drawback I could imagine from the temporal concatenation is the strong reliance of every generation frame on the input conditions (or condition shortcuts), which may lead to less dynamics in the output videos. VBench provides metrics beyond just consistency (including dynamic degree), but they are not reported in the tables.

(-) Ablation study on the number of buffer frames. I like the design of buffer frames and that the noise schedule of them is compared carefully. But it seems to me that the paper should also come with certain takeaways or conclusions based on quantitative evaluations (beyond qualitative figures in the appendix) on how many buffer frames to choose. There are some interesting questions to ask: Is there a connection between the number of buffer frames and the number of frames to be generated? How about the connection with the number of condition images/frames?

The writing could be improved by making some of the claims more rigorous. See below points

(-) I feel that the claim of "requires no architectural changes" is a bit off, as TIC-FT indeed requires additional LoRA parameters. The essential difference between TIC-FT and baselines, such as ControlNet, is probably the fact that the former introduces a much smaller number of new parameters, and these new modules are more general and not specifically designed for a certain type of visual condition.

(-) I am unsure if the setting falls within the definition of in-context learning. Usually, the in-context demonstrations consist of several examples belonging to the same tasks. The V2V setting seems fine, but the image-to-video setting doesn't really sound like in-context learning because the input image is merely a condition used for generating the video instead of being a "demonstration" that falls into the same modality or task type as the output video frames.

---

> ### Author Rebuttal · Authors · 2025-07-30
>
> We sincerely thank the reviewer for the thoughtful and detailed feedback. Your insightful comments have helped us identify key areas to clarify and improve. We have carefully addressed each concern below.
> ***
> > (-) Small-scale evaluation set. Each task is evaluated on only 20 condition-target pairs, which may not be very representative of showing the whole picture of the performance superiority. Are there larger scale benchmarks or more standard ones that can be used for evaluation?
>
> We would like to first clarify that the 20 condition-target pairs were **used for training** to demonstrate our method's few-shot efficiency. The quantitative evaluation reported in our initial submission was conducted using standard and diverse benchmarks **including VBench [1], GPT-4o, and Perceptual similarity scores**, with 100 unseen video samples per task. We did mention it in the manuscript, but will improve the camera-ready version for greater clarity!
>
> To further strengthen our evaluation and address the reviewer’s concern, we have benchmarked our method on Video-Bench [2], a newly proposed large-scale benchmark recently presented at CVPR 2025.  Video-Bench offers human-aligned evaluation metrics powered by multimodal LLMs, which demonstrate stronger correlation with human judgment compared to previous benchmarks such as VBench.
>
> The results on Video-Bench are presented below:
>
> **Table: Evaluation on Video-Bench**
>
> | method | aesthetic quality | imaging quality | motion effects | temporal consistency |
> | --- | --- | --- | --- | --- |
> | ControlNet | 3.60 | 3.58 | 2.70 | 4.75 |
> | Fun-pose | 3.94 | 4.42 | 3.05 | 4.82 |
> | TIC-FT-Replace | 3.92 | 4.29 | 3.07 | 4.80 |
> | TIC-FT (w/o Buffer) | 3.93 | 4.15 | 3.06 | 4.82 |
> | **TIC-FT** | **3.96** | **4.52** | **3.09** | **4.83** |
>
>
> As the tables show, our **TIC-FT** achieves competitive or state-of-the-art performance across most metrics on this large-scale, human-aligned benchmark (notably with less compute and better condition reflectance).
>
> We believe these additional results on a large-scale, standardized benchmark effectively demonstrate the generalizability and robust performance of our method.
>
> [1] Huang et al., Vbench: Comprehensive benchmark suite for video generative models
>
> [2] H. Han et al., “Video‑Bench: Human‑Aligned Video Generation Benchmark,” in Proceedings of the IEEE/CVF Conference on Computer Vision and Pattern Recognition (CVPR), 2025.
> ***
> > (-) Metrics: One drawback I could imagine from the temporal concatenation is the strong reliance of every generation frame on the input conditions (or condition shortcuts), which may lead to less dynamics in the output videos. VBench provides metrics beyond just consistency (including dynamic degree), but they are not reported in the tables.
>
> We thank the reviewer for raising this important point. We provide the VBench Dynamic Degree metrics in the tables below for both I2V and V2V tasks, and show that our method substantially improve dynamic degree compared to other methods. While pixel-aligned conditioning methods such as ControlNet are known to degrade dynamic degree, temporal concatenation method (Ours) is less susceptible to this issue. Additionally, our fine-tuning approach leverages the inherent data distribution more effectively than alternative methods, limiting such degradation.
>
> **Table: I2V Evaluation on VBench Dynamic Degree**
> | Method               | Dynamic Degree |
> |----------------------|----------------|
> | ControlNet           | 0.26           |
> | Fun-pose             | 0.64           |
> | **TIC-FT**               | **0.73**           |
> | *Training Data*        | *0.83*           |
>
> **Table: V2V Evaluation on VBench Dynamic Degree**
> | Method               | Dynamic Degree |
> |----------------------|----------------|
> | ControlNet           | 0.21           |
> | Fun-pose             | 0.69           |
> | **TIC-FT**           | **0.71**           |
> | *Training Data*        | *0.77*           |
>
> We further report the Dynamic Degree Discrepancy, computed as the difference between the Dynamic Degree of the Training Data and that of each generated video. (The best model should have the smallest absolute value of the discrepancy). The results show our method closely matches dynamic degree of the training data.
>
> **Table: I2V Evaluation on VBench Dynamic Degree Discrepancy**
>
> | method | Dynamic Degree Discrepancy |
> | --- | --- |
> | ControlNet | 0.57  |
> | Fun-pose | 0.19  |
> | SIC-FT-Replace | 0.22 |
> | TIC-FT-Replace | **0.10** |
> | TIC-FT (w/o Buffer) | 0.14 |
> | **TIC-FT** | **0.10** |
>
> **Table: V2V Evaluation on VBench Dynamic Degree Discrepancy**
>
> | method | Dynamic Degree Discrepancy |
> | --- | --- |
> | ControlNet | 0.56 |
> | Fun-pose | 0.08 |
> | SIC-FT-Replace | -0.10 |
> | TIC-FT-Replace | -0.19 |
> | TIC-FT (w/o Buffer) | -0.17 |
> | **TIC-FT** | **0.06** |
>
>
> > (-) Ablation study on the number of buffer frames.
>
> We have performed additional quantitative analysis to derive insights on how the number of condition and buffer frames impacts the final output.
>
> **1. Impact of the Number of Condition Frames**
>
> We found that increasing the number of condition frames in I2V tasks lead to a marginal but consistent improvement in reflecting the condition's visual details. As shown in the table below, most of the similarity metrics show a slight upward trend with more condition frames.
>
> | # Condition | CLIP-I | LPIPS (↓) | SSIM | DINO |
> | --- | --- | --- | --- | --- |
> | 1 | 0.8329 | 0.7493 | 0.5917 | 0.5530 |
> | 3 | 0.8332 | 0.7390 | 0.5918 | 0.5531 |
> | 6 | 0.8371 | 0.7360 | 0.6078 | 0.5606 |
> | 9 | **0.8396** | **0.7346** | **0.6083** | **0.5643** |
>
> **2. Impact of the Number of Buffer Frames**
>
> We observed that increasing the number of buffer frames enhances video dynamics in I2V tasks. This occurs because additional buffer frames create a temporal gap between the static input and target videos, leading to disentanglement between the two.
>
> | # Buffer | Dynamic Degree |
> | --- | --- |
> | 3 | 0.73 |
> | 6 | 0.77 |
> | 9 | **0.82** |
>
> We also observed that, as the model further disentangles the target frames from the condition frame, the influence of the condition on the generated target video tends to diminish, which is intuitive.
>
> | # Buffer | CLIP-I   | LPIPS  | SSIM   | DINO   |
> |----------|--------|--------|--------|--------|
> | 3        |**0.7812** | **0.6112** | **0.4130** | **0.4259** |
> | 6        |0.7695 | 0.6121 | 0.4127 | 0.4170 |
> | 9        | 0.7544 | 0.6232 | 0.3952 | 0.4152 |
>
> In conclusion, increasing the number of condition frames results in stronger reflection of conditioning on the generated frames, though the improvement is marginal. Additionally, more buffer frames lead to disentanglement between condition and target frames. In I2V generation, since condition frames are static, additional buffer frames increase the dynamics of the generated video, while at the cost of reduced condition reflection.
>
> Given that the influence of condition frame count is marginal, it is optimal to use the minimum number of condition frames, as additional frames require more memory and computation. We found that 1 condition frame is optimal for I2V tasks. For buffer frames, we determined that 3 buffer frames provide the optimal balance between memory usage, condition reflection, and dynamic degree.
>
> ***
> > (-) I feel that the claim of "requires no architectural changes" is a bit off, as TIC-FT indeed requires additional LoRA parameters.
>
> We agree with the reviewer that our original claim of "no architectural changes" is imprecise, as TIC-FT does introduce new LoRA parameters. We thank the reviewer for this point and will revise our manuscript with a more accurate description.
>
> Our original intention was to highlight that, while our method introduces new trainable LoRA parameters, it does not alter the model's high-level architecture or its computational graph. (LoRA modifies the behavior within existing operational blocks rather than adding new, parallel network branches.)
>
> (As the reviewer suggested) This preservation of the high-level architecture is what makes our TIC-FT framework inherently modality-agnostic and general-purpose, as no specialized modules are needed to handle different model architectures.
>
>
> ***
> > (-) I am unsure if the setting falls within the definition of in-context learning.
>
> To clarify, our paper does not claim to implement in-context learning (ICL) in the traditional sense as defined in the context of LLMs.
> Instead, we define our work as "in-context finetuning", a related but distinct paradigm adapted for diffusion models. It is an efficient fine-tuning technique inspired from ICL, but involves finetuning, that leverages spatial or—in our case—temporal concatenation to facilitate information sharing across a sequence. Our novelty lies in extending spatial In-Context finetuning used for image generation to video generation, with Temporal In-Context Fine-Tuning (TIC-FT), along with techniques like adding buffer frames.
> ***
> We sincerely appreciate your time and constructive feedback. We believe the additional results, clarifications, and analyses presented above effectively address your concerns and look forward to further discussion.

---

### Note · Authors · 2025-08-15

Dear Reviewers and Area Chairs,

We sincerely thank Reviewers QrUT, ngnG, 9QKt, and juYd for their thoughtful and constructive feedback.
Our work introduces Temporal In-Context Fine-Tuning (TIC-FT), a method to finetune pretrained video diffusion models for conditional video generation.

It minimizes the **distribution gap between pretraining and finetuning for conditional video diffusion** and achieves strong performance with **only a small number of training samples.** It further supports **versatile conditioning** for diverse image- and video-conditioned tasks, and has been validated on representative applications such as reference-to-video generation, motion transfer, and style transfer.

During the rebuttal period, we addressed the reviewers’ concerns and questions as follows, which will be incorporated into the revised paper:

- Clarified that evaluations used 100 unseen videos per task and added Video-Bench results, showing competitive or SOTA performance across quality and temporal metrics.
- Provided ablation insights on condition and buffer frame counts, showing that more condition frames improve condition adherence, while more buffer frames enhance motion at some cost to adherence.
- Explained that ControlNet and Fun-pose (as a concatenation-based approach) were chosen as representative baselines of leading paradigms, with TIC-FT outperforming both while using less GPU memory.

We appreciate the reviewers’ acknowledgment that the main concerns have been addressed. We are grateful for their feedback, which has helped us refine and improve the work in meaningful ways.

---

### Decision · Program_Chairs · 2025-09-17

**Decision:**

Accept (poster)

**Comment:**

Summary

This paper introduces Temporal In-Context Fine-Tuning (TIC-FT), a method for adapting pretrained video diffusion models to diverse conditional video generation tasks (e.g., image-to-video, video-to-video). The key innovation is to concatenate condition and target frames along the temporal axis, inserting buffer frames with progressively increasing noise levels to smooth the transition and align with the pretrained model’s temporal dynamics. TIC-FT requires no architectural changes (beyond LoRA parameters), is highly parameter- and compute-efficient, and achieves strong performance with as few as 10–30 training samples. The method is validated on large-scale models (CogVideoX-5B, Wan-14B) and a range of tasks, outperforming established baselines in condition fidelity, visual quality, and efficiency.

Strengths
- Novelty & Simplicity: The temporal concatenation and buffer frame mechanism is a creative, elegant solution for conditional video generation, enabling versatile control without architectural modifications.
- Efficiency: TIC-FT is highly parameter- and memory-efficient, requiring minimal training data and compute, making it accessible for low-resource users.
- Empirical Validation: Extensive experiments on multiple tasks and large-scale models demonstrate superior performance over strong baselines (ControlNet/ControlVideo, Fun-pose), including on human-aligned benchmarks (VBench).
- Clarity: The paper is well-written, with clear methodological exposition and thorough ablation studies.
- Extensibility: The approach generalizes across diverse conditional tasks and is compatible with various pretrained backbones.

Weaknesses
- Baseline Coverage: While ControlNet and Fun-pose are relevant baselines, some reviewers noted the absence of comparisons to other recent video-specific adapters (e.g., VidControl). Authors clarified ControlNet is functionally equivalent to ControlVideo
- Temporal Metrics: Initial evaluation focused on FVD/CLIP; reviewers requested more fine-grained motion metrics (e.g., dynamic degree, temporal consistency). Authors addressed this with additional results on VBench, showing competitive or SOTA performance.
- Long-Sequence Scalability: The method is currently limited to ~10-second videos due to memory constraints. While this is a common limitation, future work on longer sequences would be valuable.
- Precise Frame-Level Control: TIC-FT is less suited for dense, pixel-aligned conditioning (e.g., per-frame depth maps) compared to ControlNet-style methods.
- Definition of "In-Context": Some reviewers questioned the use of "in-context learning" terminology, as the method is more akin to in-context fine-tuning than classical ICL.


Overall, the paper presents a technically solid and positively received by all reviewers. The method is simple and effective, with strong empirical results and broad applicability. Most reviewer concerns were addressed convincingly in the rebuttal, including expanded evaluation, clarification of baselines, and detailed ablation studies. ACs concur with reviewer consensus. Authors must incorporate reviewer feedback in the final paper.